# MFIT 1.0.0: Multiflow inversion of tracer breakthrough curves in fractured and karst aquifers

Jacques Bodin[1]

[1]Université de Poitiers, CNRS, UMR 7285 IC2MP, 40 Avenue du Recteur Pineau, 86022 Poitiers Cedex, France

*Correspondence to*: Jacques Bodin (jacques.bodin@univ-poitiers.fr)

**Abstract.** More than half of the Earth's population depends largely or entirely on fractured or karst aquifers for their drinking water supply. Both the characterization and modeling of these groundwater reservoirs are therefore of worldwide concern. Artificial tracer testing is a widely used method for the characterization of solute (including contaminant) transport in groundwater. Tracer experiments consist of a two-step procedure: 1) introducing a conservative tracer-labeled solution into an
aquifer, usually through a sinkhole or a well, and 2) measuring the concentration breakthrough curve (BTC) response(s) at one or several downstream monitoring locations, usually spring(s) or pumping well(s). However, the modeling and interpretation of tracer test responses can be a challenging task in some cases, notably when the BTCs exhibit multiple local peaks and/or extensive backward tailing. MFIT is a new open-source, Windows-based computer package for the analytical modeling of tracer BTCs. This software integrates four transport models that are all capable of simulating single- or multiple-peak and/or
heavy-tailed BTCs. The four transport models are encapsulated in a general multiflow modeling framework, which assumes that the spatial heterogeneity of an aquifer can be approximated by a combination of independent one-dimensional channels. Two of the MFIT transport models are believed to be new, as they combine the multiflow approach and the double-porosity concept, which is applied at the scale of the individual channels. Another salient feature of MFIT is its compatibility and interface with the advanced optimization tools of the PEST suite of programs. Hence, MFIT is the first BTC fitting tool that
allows regularized inversion and nonlinear analysis of the postcalibration uncertainty of model parameters.

## 1 Introduction

Artificial tracer testing is one of the most valuable methods for the characterization of flow and solute transport in groundwater. Tracer experiments consist of a two-step procedure: 1) introducing a known mass of a tracer species into an aquifer, usually through a sinkhole or well, and 2) measuring the concentration breakthrough curve (BTC) response(s) at one or several
downstream monitoring locations, usually spring(s) or pumping well(s). The analysis of a tracer BTC is best done by fitting a model-computed time-concentration curve to the measured values. Although spatially distributed numerical models (e.g., MODFLOW/MT3DMS or FEFLOW) can be used for this purpose, simpler (i.e., spatially lumped) models are generally used, at least in the early stages of tracer studies, either because of time constraints or because of a lack of model input data. A number of computer codes for BTC fitting have been developed in recent decades: CATTI (Sauty et al., 1992), TRACI (Käss,

1998, 2004), OTIS (Runkel, 1998), STANMOD (van Genuchten et al., 2012), TRAC (Gutierrez et al., 2013), OM-MADE (Tinet et al., 2019), and OptSFDM (Gharasoo et al., 2019). Note that STANMOD integrates a number of former codes, including the widely used CXTFIT code developed by Parker and van Genuchten (1984) and Toride et al. (1999). Despite the range of possibilities offered by these programs, the fitting and interpretation of tracer BTCs remains a challenging task in some cases, notably for BTCs exhibiting multiple local peaks and extensive backward tailing. Such BTC shapes, which fall in the general category of non-Fickian (or anomalous) transport (Berkowitz et al., 2006; Neuman and Tartakovsky, 2009), are frequently observed in fractured and karst aquifers (Tsang and Neretnieks, 1998; Streetly et al., 2002; Massei et al., 2006; Loefgren et al., 2007; Goldscheider et al., 2008; Field and Leij, 2012; Bertrand et al., 2015; Yang et al., 2019).

To the best of the author's knowledge, only the TRACI and OM-MADE programs are able to simulate multimodal BTCs. Unfortunately, these two programs suffer from some limitations both in terms of ease of use and with respect to their modeling/calibration capabilities. For instance, the TRACI software has not been maintained since 2004 and can only be used on physical or virtual computers running Windows operating system versions from Windows 98 to Windows 7. Another drawback of TRACI is the inability of the inversion (automated calibration) algorithm included in the software to handle multimodal BTCs. Each local concentration peak must be sequentially fitted through a manual (trial-and-error) calibration procedure. The OM-MADE program was written as a Python script and has neither a graphical user interface (GUI) nor inverse modeling functionality. The purpose of this paper is to present a new open-source GUI-based software, named MFIT, that aims to help in the interpretation of single- or multiple-peak and/or heavy-tailed BTCs. MFIT stands for "Multi-Flow Inversion of Tracer breakthrough curves". The MFIT software integrates four transport models that can be tested against field and laboratory tracer BTCs with the assistance of the PEST automated calibration and uncertainty analysis routines (Doherty, 2019a). In its current version, the scope of the software is limited to tracer tests involving nonreactive tracer species and performed in steady flow conditions. These assumptions are maintained throughout the paper.

The remainder of this paper is organized as follows. Section 2 discusses the possible origins of multiple peaks and long tails in tracer BTCs and presents the conceptual and mathematical framework of the transport models integrated in MFIT. The code implementation and coupling with PEST for automated BTC fitting are discussed in section 3. In section 4, the accuracy of MFIT-computed BTCs is verified against CXTFIT and TRACI simulations for five test cases. An additional test is presented in section 5 to assess the reliability of a new multistart method that was specifically developed to improve the automatic optimization of the model parameters. In section 6, we illustrate the use of the software by analyzing tracer BTCs obtained in the karst aquifer of the Hydrogeological Experimental Site (HES) in Poitiers, France. The summary and conclusions are presented in section 7. A number of acronyms, model abbreviations, and model parameters are employed throughout this paper. Two glossaries, Tables A1 and A2, are provided in Appendix A for easy reference.

## 2 Multimodal and heavy-tailed BTCs: causes and modeling

Under the already mentioned assumptions (nonreactive tracer, steady-state flow), multimodal BTCs unequivocally indicate that a number of tracer-plume splittings occurred somewhere between the injection site and the monitoring point. Although injection artifacts may be involved in some cases, see, e.g., Guvanasen and Guvanasen (1987), tracer splitting most commonly originates from the spreading (transverse dispersion) of the solute into areas of contrasting flow velocities; see, e.g., Moreno and Tsang (1991), Siirila-Woodburn et al. (2015), and Boon et al. (2017). More precisely, assuming a single-pulse tracer injection signal, multimodal BTCs reflect a three-step process: 1) tracer spreading into different flowing or nonflowing aquifer subdomains characterized by different transit/residence times, 2) tracer motion within each subdomain with little or no exchange between the different subdomains, and 3) convergence (mixing) of the subtracer fluxes somewhere upstream from, or at, the monitoring point. The different models that have been proposed in the literature for simulating multimodal tracer BTCs share a common "multiflow" approach initially proposed by Zuber (1974) for the modeling of layered aquifers. In this approach, which is depicted in Fig. 1, the flow system is described as a juxtaposition of a number of one-dimensional (1-D) channels that are connected by a single common diverging (splitting) node at the entrance to the system and a single common converging (mixing) node at the outlet.

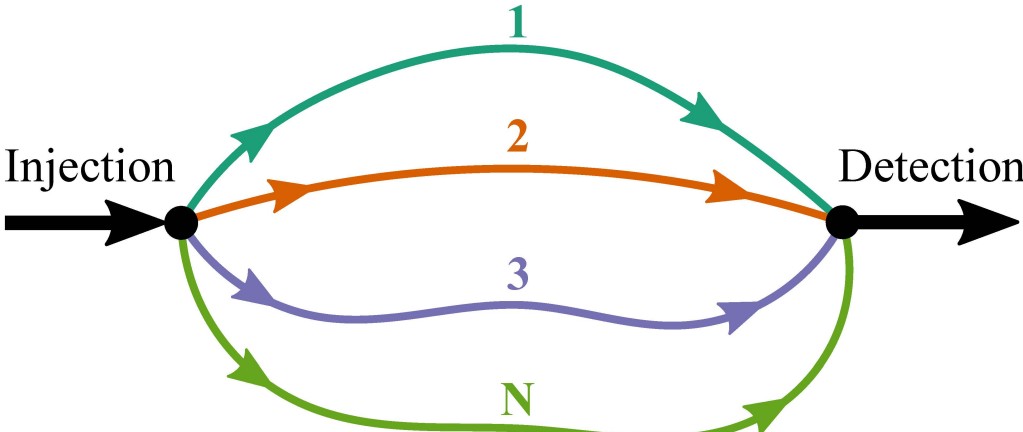

**Figure 1.** Conceptual sketch of the (generic) multiflow modeling approach, modified from Leibundgut et al. (2009)

In the multi-dispersion model (MDM) proposed by Maloszewski et al. (1992) and implemented in the TRACI software, the transport along each channel is assumed to obey the one-dimensional (1-D) advection-dispersion equation (ADE), and no mass exchange is allowed between different channels. In the dual-advection-dispersion equation (DADE) model proposed by Field and Leij (2012), only two channels are considered. The tracer is transported by advection and dispersion along each channel, and mass exchanges between the two domains are possible. These exchanges are assumed to be governed by a first-order process. The transport model implemented in the OM-MADE code can be viewed as a generalization of the DADE model, where (i) a larger number of channels can be used, (ii) each channel can be discretized to a number of subelements with

different hydraulic and transport properties, and (iii) some channels can be specified as nonflowing (stagnant) water volumes. Mass exchanges between the different channels (either flowing or nonflowing) are likewise modeled as a first-order process.

As pointed out above, the production of a multimodal BTC requires little or no exchange between the subtransport domains; otherwise, the mixing of the mass fluxes would rehomogenize the subtracer plumes. In accordance with this principle, small exchange coefficient values must be used in the DADE and OM-MADE models for simulating multimodal BTCs, and this approach makes these models converge toward the MDM.

The interpretation of the long-tail behavior of a BTC may be more difficult than that of multiple peaks, as different mechanisms

can be involved. The possible sources of extensive BTC tailing can be listed as follows: (i) tracer retention/decaying boundary condition at the injection site; (ii) tracer splitting into well-separated flow paths and then downstream convergence/mixing/overlapping of the individual pathway responses; and (iii) mass exchanges between flow domains characterized by different transit/residence times. The above-listed processes are referred to below as "injection decay", "multiflow overlapping", and "multiflow exchanges", respectively. The MDM can simulate long-tailed BTCs as a result of

multiflow overlapping. Multiflow exchanges are the core of the DADE model, and both multiflow overlapping and multiflow exchanges can be combined in the OM-MADE model. A number of other models have been proposed in the literature for simulating unimodal long-tailed BTCs; see, e.g., reviews in Bodin et al. (2003b), Neuman and Tartakovsky (2009), Zhang et al. (2009), Dentz et al. (2011) and examples of recent works in Field and Leij (2014) and Labat and Mangin (2015). The two most commonly used models for the analysis of artificial tracer tests are the two-region nonequilibrium (2RNE) model of

Toride et al. (1993), implemented in the CXTFIT code, and the single-fracture dispersion model (SFDM) of Maloszewski and Zuber (1990), implemented in TRACI and OptSFDM software. Both the 2RNE model and SFDM assume mass exchange between a single mobile (flowing) domain and a single immobile domain. A key distinction between the 2RNE model and SFDM is the formulation of mass exchange, which is described as a first-order process in the 2RNE model (as in the DADE and OM-MADE models) and as a second-order (diffusion) process in the SFDM.

As already noted, multimodal and long-tailed BTCs are typical of tracer tests performed in fractured and karst aquifers. A common feature of both aquifer types is the existence of low hydraulic resistance pathways provided by the fractures and karst conduits (Tsang and Neretnieks, 1998; Worthington and Ford, 2009). A generic multiflow modeling approach is therefore intuitively appealing for the interpretation of tracer tests in fractured and karst aquifers. Of course, the actual (and generally unknown) geometry of the discrete flow network experienced by the tracer is likely more complex than that depicted in Fig.

1. The channels are therefore not assumed to represent individual fractures or karst conduits but are lumped submodels of the main flow routes used by the tracer through the fractures/karst conduit network. The four transport models integrated in the MFIT software are based on the multiflow approach. The first model is a reimplementation of the MDM. The second model is a variant of the MDM that assumes an exponentially decaying injection of the tracer concentration at the inlet of the flow system. In the third and fourth models, the double-porosity concept (2RNE model and SFDM) is applied at the scale of the

individual channels. It is unclear whether this idea of combining multiflow and double-porosity systems is new. In the TRACI software, it is technically possible to fit a series of SFDM curves to a multimodal tracer BTC and then calculate the mean

combined model curve, but to the best of the author's knowledge, this method has never been discussed or applied in the literature. A possible reason is the increasing number of fitting parameters, which makes the inverse problem more complicated. Among the challenges related to the inversion of a multiflow model is the inherent problem of nonuniqueness
(or equifinality). A variety of parameter sets can yield nearly identical simulated BTCs because the change in the value of a parameter of a given channel can be compensated by modifying at least one other parameter that pertains to this same channel or the parameters of the other channels. This nonuniqueness causes the inverse problem to be ill-posed in the sense of Hadamard (1902) and requires the use of advanced optimization methods, such as regularization, to make the inverse problem tractable (Tikhonov and Arsenin, 1977; Moore and Doherty, 2006; Zhou et al., 2014).

In this article, the combination of multiflow and double-porosity systems is referred to as the multi-double porosity (MDP) approach. The immobile domain that is assigned to each flow channel is assumed to describe the porous rock matrix in contact with the fractures/karst conduits and/or any other stagnant water zones (e.g., pool volumes) adjacent to the main tracer pathways. For each of the four MFIT models, the channels are assumed to be independent of each other, i.e., no mass exchange is allowed between the channels. Actually, this assumption is mathematically convenient rather than physically motivated. As
already indicated, the channels are abstractions of the real main tracer pathways, which may cross (and therefore exchange between) each other between the injection site and the monitoring point. Assuming fully separated channels allows analytical modeling of mass fluxes in the multiflow system, and this approach makes the inversion of model parameters computationally more efficient (see discussion in section 3).

The governing equations of the transport models are given as follows. The concentration at the outlet of a multiflow system as
depicted in Fig. 1 can be calculated from the mass flux balance as follows:

$$QC = \sum_{j=1}^{N} Q_j C_j \qquad (1)$$

where $Q(\text{L}^3\text{T}^{-1})$ is the total system flow rate; $C(\text{ML}^{-3})$ is the outflow concentration; $N$ is the number of flow channels; the subscript $j$ denotes the flow channel index; and $Q_j(\text{L}^3\text{T}^{-1})$ and $C_j(\text{ML}^{-3})$ are the flow rate and concentration in the $j$-th channel, respectively.

The mathematical equations that have been used by Maloszewski et al. (1992) in the MDM to describe the solute transport in each flow channel are the 1-D ADE as follows:

$$\frac{\partial c_j}{\partial t} = -u_j \frac{\partial c_j}{\partial x_j} + D_j \frac{\partial^2 c_j}{\partial x_j^2} \qquad (2)$$

and its analytical solution for the case of an instantaneous solute injection in a semi-infinite medium with both injection and detection in flux, which is expressed as (Kreft and Zuber, 1978)

$$C_j = \frac{m_j}{2Q_j T_{0j} \sqrt{\frac{\pi}{Pe_j}\left(\frac{t}{T_{0j}}\right)^3}} exp\left(-\frac{Pe_j T_{0j}}{4t}\left(1 - \frac{t}{T_{0j}}\right)^2\right) \qquad (3)$$

where $t$(T) is the time variable; $x_j$(L) is the spatial coordinate along the $j$-th flow channel; $u_j$(LT$^{-1}$) and $D_j$(L$^2$T$^{-1}$) are the advection velocity and the dispersion coefficient, respectively; $m_j$(M) is the part of the solute mass that flows through the $j$-th channel; and $T_{0j}$(T) and $Pe_j$(−) are the mean transit time and Peclet number, respectively, which are expressed as

$$T_{0j} = \frac{L_j}{u_j} \tag{4}$$

$$Pe_j = \frac{u_j L_j}{D_j} \tag{5}$$

where $L_j$(L) is the length of the $j$-th pathway. Substituting Eq. (3) into Eq. (1) yields:

$$C = \frac{1}{Q} \sum_{j=1}^{N} \frac{m_j}{2T_{0j}\sqrt{\frac{\pi}{Pe_j}\left(\frac{t}{T_{0j}}\right)^3}} exp\left(-\frac{Pe_j T_{0j}}{4t}\left(1 - \frac{t}{T_{0j}}\right)^2\right) \tag{6}$$

The calibration of Eq. (6) against a tracer test BTC requires determination of the total system flow rate $Q$; the number $N$ of flow channels; and for each flow channel, the values of $m_j$, $T_{0j}$ and $Pe_j$. In this work, we generalize the above-described method by considering alternative models for the transport in individual channels and substituting the related analytical expressions of $C_j$ into Eq. (1). The analytical transport models that are considered are (i) the solution of Eq. (2) for the case of a decaying injection boundary condition, (ii) the SFDM, and (iii) the 2RNE model.

The analytical solution of Eq. (2) for the case of a decaying injection boundary condition $C_j(x_j = 0, t) = C_0 \, exp(-\lambda_j t)$ was derived by Marino (1974) and can be written in the following form:

$$C_j = \frac{C_0}{2}\left[\begin{array}{l} erfc\left(\left(1 + \frac{\gamma_j t}{T_{0j}}\right)\sqrt{\frac{Pe_j T_{0j}}{4t}}\right)exp\left(\frac{\gamma_j Pe_j}{2}\right) \\ + erfc\left(\left(1 - \frac{\gamma_j t}{T_{0j}}\right)\sqrt{\frac{Pe_j T_{0j}}{4t}}\right)exp\left(\frac{-\gamma_j Pe_j}{2}\right)\end{array}\right]exp\left(\frac{Pe_j}{2}\left(1 - \frac{(1-\gamma_j^2)t}{2T_{0j}}\right)\right) \tag{7}$$

where $C_0$(ML$^{-3}$) is the initial (maximum) injection concentration at the inflow boundary, $\lambda_j$(T$^{-1}$) is the time decay constant, and

$$\gamma_j = \sqrt{1 - \frac{4D_j\lambda_j}{u_j^2}} \tag{8}$$

The SFDM developed by Maloszewski and Zuber (1990) describes solute transport in a double-porosity fracture-matrix system. The considered transport mechanisms are advection-dispersion in the fracture and diffusion in the surrounding rock matrix. The fracture is idealized as a parallel-plate channel, and the matrix diffusion is assumed to be unlimited, i.e., not influenced by the fluxes from other fractures. The transport equations can be written as follows:

$$\frac{\partial c_j}{\partial t} = -u_j\frac{\partial c_j}{\partial x_j} + D_j\frac{\partial^2 c_j}{\partial x_j^2} + \frac{\theta_{pj}D_{pj}}{b_j}\frac{\partial c_{pj}}{\partial y_j}\bigg|_{y_j=b_j} \quad \text{for } 0 \le y_j \le b_j \tag{9}$$

$$\frac{\partial C_{pj}}{\partial t} = D_{pj} \frac{\partial^2 C_{pj}}{\partial y_j^2} \text{ for } b_j \leq y_j \leq \infty \tag{10}$$

where $C_j$(ML$^{-3}$) and $C_{pj}$(ML$^{-3}$) are the solute concentrations in the flow channel and in the rock matrix, respectively; $\theta_{pj}(-)$ is the matrix porosity; $D_{pj}$(L$^2$T$^{-1}$) is the molecular diffusion coefficient in the matrix; $b_j$(L) is the half-aperture of the flow channel; and $y_j$(L) is the spatial coordinate perpendicular to the channel extension. The solution to Eqs. (9) and (10) for the case of an instantaneous injection is

$$C_j = \frac{m_j \beta_j \sqrt{Pe_j T_{0j}}}{2\pi Q_j} \int_0^t \frac{exp\left(-\frac{Pe_j(T_{0j}-\xi)^2}{4T_{0j}\xi} - \frac{\beta_j^2 \xi^2}{t-\xi}\right)}{\sqrt{\xi(t-\xi)^3}} d\xi \tag{11}$$

where $\xi$(T) is the integration variable and $\beta_j$(T$^{-1/2}$) is the so-called diffusion parameter defined as:

$$\beta_j = \frac{\theta_{pj}\sqrt{D_{pj}}}{2b_j} \tag{12}$$

Coats and Smith (1964) proposed a different mathematical formulation of solute mass exchange between flowing and stagnant water regions in double-porosity media, which is well known in the literature either as the mobile-immobile (MIM) model or the 2RNE model as

$$\theta_j \frac{\partial C_j}{\partial t} + \theta_{imj} \frac{\partial C_{imj}}{\partial t} = \theta_j \left(-u_j \frac{\partial C_j}{\partial x_j} + D_j \frac{\partial^2 C_j}{\partial x_j^2}\right) \tag{13}$$

$$\theta_{imj} \frac{\partial C_{imj}}{\partial t} = \alpha_j \left(C_j - C_{imj}\right) \tag{14}$$

where $\theta_j(-)$ and $\theta_{imj}(-)$ are the mobile and immobile volumetric water contents, respectively, $C_{imj}$(ML$^{-3}$) is the concentration in the immobile domain, and $\alpha_j$(T$^{-1}$) is a first-order mass transfer coefficient. The two main differences with respect to the SFDM are (i) the dual-domain formulation of the problem (mobile and immobile regions are assumed to coexist at each point in space, and this assumption differs from the parallel-plate channel geometry in the SFDM) and that (ii) the solute mass exchange between mobile and immobile domains is assumed to be governed by a first-order process, whereas the SFDM refers to the second-order diffusion Eq. (10). Building on a general set of analytical solutions developed by Toride et al. (1993), the solution of the 2RNE model for the case of an instantaneous injection can be written as follows:

$$C_j = \frac{m_j}{Q_j} \left[ \frac{1}{T_{0j}\sqrt{\frac{4\pi}{Pe_j}\left(\frac{t}{T_{0j}}\right)^3}} exp\left( -\frac{Pe_j\left(1-\frac{t}{T_{0j}}\right)^2}{4\frac{t}{T_{0j}}} - \omega_j L_j \frac{t}{T_{0j}} \right) \right.$$
$$\left. + \frac{\omega_j \psi_j}{T_{0j}}\sqrt{\frac{L_j^3 Pe_j}{4\pi(1-\psi_j)}} \int_0^{\psi_j L_j \frac{t}{T_{0j}}} \frac{1}{\tau\sqrt{\psi_j L_j \frac{t}{T_{0j}} - \tau}} exp\left( -\frac{Pe_j(\psi_j L_j - \tau)^2}{4\psi_j L_j \tau} - \frac{\omega_j \tau}{\psi_j} - \frac{\omega_j\left(\psi_j L_j \frac{t}{T_{0j}} - \tau\right)}{1-\psi_j} \right) I_1\left( 2\omega_j\sqrt{\frac{\tau\left(\psi_j L_j \frac{t}{T_{0j}} - \tau\right)}{\psi_j(1-\psi_j)}} \right) d\tau \right] \tag{15}$$

where $I_1$ is the modified Bessel function of the first kind, $\tau(L)$ is the integration variable, and

$$\psi_j = \frac{\theta_j}{\theta_j + \theta_{imj}} \tag{16}$$

$$\omega_j = \frac{\alpha_j}{\theta_j u_j} \tag{17}$$

It is notable that when $\psi_j=1$ and $\omega_j=0$, Eq. (15) simplifies to Eq. (3). Table 1 summarizes the parameters of the four MFIT transport models.

**Table 1.** Parameters of the transport models integrated in the MFIT software. The subscript $j$ denotes a parameter that must be defined for each flow channel. The parameters without this subscript are common to all channels.

| Model | Parameters | Maximum number of calibration parameters for an $n$-channel solution |
|---|---|---|
| MDMi (ADE, instantaneous injection) | $Q$, $m_j$, $T_{0j}$, $Pe_j$ | $3n+1$ |
| MDMed (ADE, exponentially decaying injection) | $C_0$, $Q_j/Q$, $T_{0j}$, $Pe_j$, $\gamma_j$ | $4n+1$ |
| MDP-SFDM | $Q$, $m_j$, $T_{0j}$, $Pe_j$, $\beta_j$ | $4n+1$ |
| MDP-2RNE | $Q$, $m_j$, $L_j$, $T_{0j}$, $Pe_j$, $\psi_j$, $\omega_j$ | $6n+1$ |

## 3 Code implementation and inversion

The four analytical models described in the previous section have been implemented in C++ and compiled as executable Windows programs named MDMi.exe (for MDM, instantaneous injection), MDMed.exe (for MDM, exponentially decaying

injection), MDP_SFDM.exe, and MDP_2RNE.exe. The code and executable files are freely available on the public Zenodo repository https://doi.org/10.5281/zenodo.3470751. In the MDP_SFDM and MDP_2RNE programs, the numerical evaluation of the integrals in Eqs. (11) and (15) is performed using the QAG adaptive integration routine from the GNU Scientific Library with a 61-point Gauss-Kronrod rule and a relative error convergence criterion of $10^{-2}$. These four programs can be run as console applications to solve a direct (forward) problem, i.e., computing a series of time-concentration values for a given set of model parameters. Both the input and output files are in ASCII format and can be edited with any text editor program for pre-/postprocessing. A convenient alternative is to use the MFIT software as a GUI for these applications. The MFIT software has been developed using the C++ Builder environment (Embarcadero RAD Studio 10.1 Berlin) and provides a GUI for (i) importation and graphic visualization of user-provided BTC data; (ii) parameterization, direct running, and graphical output of the analytical transport models; (iii) inversion (automatic calibration) of model parameters for optimal curve fitting; and (iv) assessment of the uncertainty of calibrated parameter values.

The optimization and uncertainty analysis of the model parameters for a given number of flow channels are carried out using PEST routines (Doherty, 2019a, 2019b). The influence of the number of channels on the model fitting performance can be analyzed once a series of calibrations has been performed for a variety of channel numbers, as illustrated below. PEST is a public domain model-independent program suite that has been widely used over the past two decades, notably in the field of surface and subsurface hydrology (e.g., Long, 2015; Woodward et al., 2016; Gaudard et al., 2017; Wang et al., 2019). The theoretical framework and full range of capabilities of the PEST software are well documented (Doherty et al., 2010; Doherty, 2015, 2019a, 2019b) and are not repeated here. Only the concepts and methods that were deemed to be the most relevant to the multiflow modeling approach and that have been made accessible through the MFIT GUI software are briefly reviewed below.

PEST is based on a gradient optimization method and, as such, requires the derivatives of model outputs with respect to the adjustable model parameters to be calculated in each iteration for implementing the Jacobian (sensitivity) matrix. As pointed out by Doherty (2015), the accuracy of these derivative calculations is critical to the performance of the PEST optimization algorithm. In the MFIT program suite, most of the model partial derivatives are calculated analytically and externally provided to PEST. This approach ensures both the accuracy and speed of this part of the optimization process. Less straightforward partial derivative expressions were derived using MAPLE and exported as C code using the MAPLE code generation routine. The partial derivative functions were implemented in the MDMi, MDMed, MDP_SFDM, and MDP_2RNE programs and are processed during the PEST system calls to these programs by providing an optional "/d" command line argument to the program name. In a few cases, however, the partial derivatives cannot be calculated analytically, as they involve undefined limits. Such is the case for the derivatives of Eq. (15) with respect to the parameters $\psi_j$, $L_j$ and $T_{0j}$. In these cases, the partial derivatives are computed by PEST using finite differences.

The calibration of a multiflow transport model against a tracer BTC is hampered by two well-known issues in inverse modeling: (i) model nonlinearity and (ii) solution nonuniqueness. Both issues may cause numerical instabilities that can prevent the inversion algorithm from converging to the optimal solution. PEST includes two regularization methods that can be used either

individually or together to guide the optimization process. The singular value decomposition (SVD) method subtracts parameter combinations for which the tracer BTC is uninformative. The inversion is conducted on the basis of a reduced set of orthogonal linear combinations of the model parameters rather than attempting to estimate the parameters individually. The Tikhonov regularization method provides a different but complementary strategy, where the information content of the tracer BTC is supplemented with expert knowledge pertaining to the model parameters. When using Tikhonov regularization, the

objective function that is minimized by PEST is defined as the sum of two terms. The first term is the "measurement objective function" and is defined as the sum of the squared weighted differences between the real tracer BTC and the model-simulated curve. The second term is referred to as the "regularization objective function" and acts as a penalty function for deviations from some preferred parameter conditions. Two Tikhonov regularization options have been implemented in MFIT. The first option, referred to as "preferred homogeneity", promotes a solution of minimum variance for the model parameters pertaining

to the different channels. In the second option, referred to as "preferred value", the optimization process seeks the solution that is the closest to some prior estimates of the model parameters.

Unfortunately, neither SVD nor Tikhonov regularization can guarantee that the PEST optimization algorithm will converge to the global optimal solution in the parameter space. Where local minima exist in the objective function, which is the rule rather than the exception with nonlinear models, the optimization process may become trapped and fail to identify existing better

solutions (Singh et al., 2012; Espinet and Shoemaker, 2013; Abdelaziz et al., 2019). A central issue in this case is the sensitivity to initial parameter values, i.e., different initial parameter sets may lead to different optimized solutions. Global optimization methods have been proposed in the literature to overcome this issue; see, e.g., Arsenault et al. (2014) for a review and comparison of various algorithms. The PEST program suite includes two such global optimizers based on the SCE-UA method (Duan et al., 1992) and the CMA-ES method (Hansen and Ostermeier, 2001). The corresponding programs are named

SCEUA_P and CMAES_P, respectively. It must be noted, however, that global optimization methods suffer from their own drawbacks, including sensitivity to tuning parameters and low computational efficiency. An alternative strategy to improve the chances of convergence toward the global optimum with gradient-based methods is the "multistart" approach, which consists of repeating the optimization process starting from different initial parameter value sets (Skahill and Doherty, 2006; Piotrowski and Napiorkowski, 2011). Such a strategy has been implemented in the MFIT software. The key principle of the

proposed algorithm is that rather than conducting the optimization for a fixed number $N$ of channels only, a series of automatic tracer BTC fittings is performed for a decreasing number of channels ranging from $N_{max}$ to 1. The main steps of the MFIT multistart algorithm are detailed as follows:

1. The first optimization is performed by considering the maximum number of flow channels $N_{max}$. The initial transport parameters are automatically tuned by MFIT to obtain $N_{max}$ well-separated concentration peaks. For this goal, the tracer

BTC is first analyzed to determine the times $T_5$ and $T_{95}$, which are defined as

$$T_5 = max(T_{5th}, 1.1 \times T_{min}) \tag{18}$$

$$T_{95} = min(T_{95th}, 0.9 \times T_{max}) \tag{19}$$

where $T_{5th}$ and $T_{95th}$ are the earliest and latest times at which the concentration values are above and below 5 % of the maximum concentration value, respectively, and $T_{min}$ and $T_{max}$ are the minimum and maximum time values of the user-provided BTC, respectively. The mean travel times $T_{0j}$ are then uniformly distributed between the times $T_5$ and $T_{95}$. Next, the initial Peclet number $Pe_j$ of each channel is calculated as:

$$Pe_j = \left(15 \frac{N_{max}T_{0j}}{T_{95}-T_5}\right)^2 \tag{20}$$

Equation (20) is based on a semiempirical relationship between the standard deviation of travel times for transport by advection and dispersion, $\sigma_j = T_{0j}\sqrt{2/Pe_j}$ (see, e.g., Bodin et al. 2003a, Eq. 10), and the time span of the $j$-th concentration peak, which is on the order of $6\sigma_j$. The constraint of well-separated concentration peaks may be formulated as $6\sigma_j N_{max} \ll (T_{95}-T_5)$, which is verified by Eq. (20). The initial values of the other transport parameters in Eqs. (7), (11), and (15) are chosen to minimize the tailing effect due to noninstantaneous injection or solute mass exchange between flowing and stagnant water regions as follows: $\gamma_j = 0.1$, $\beta_j = 0.001$, $\psi_j = 0.9$, and $\omega_j = 0.05$.

2. Once the optimization has been performed for the $N_{max}$ channel model, the next step is to optimize the transport parameters for $N_{max}-1$ channels. The multistart optimization approach begins here as not only one but $N_{max}$ optimizations are performed in this step. The initial parameter values for the $N_{max}-1$ channels are initialized from the previously optimized $N_{max}$ channel solution by sequentially removing one of the channels. Only the solution corresponding to the lowest sum of the squared weighted differences between the tracer BTC and model-simulated curve is retained.

3. This procedure is repeated up to the single-channel solution. The total number of PEST optimizations is $N_{max}(N_{max}+1)/2$.

Calling the multistart algorithm has been made optional in MFIT, as this algorithm significantly increases the computational cost and running time of the optimization process. However, experience has shown that the multistart approach can truly improve the model fit results and can be worth the effort in many circumstances. A comparison between optimizations conducted by the PEST multistart algorithm and the global SCE-UA and CMA-ES methods was conducted in this study and is discussed in section 6.

Because of the nonuniqueness of the inverse problem, some uncertainties may be associated with the PEST-optimized model parameter values. A nonlinear analysis method has been implemented in MFIT for the assessment of postcalibration parameter uncertainty. The method is essentially similar to that described by Fang et al. (2019) and relies on the use of the PREDUNC7 and RANDPAR utilities documented in the PEST manual (Doherty, 2019b). The algorithm can be described by the following steps: 1) compute a linear approximation to the posterior parameter covariance matrix using PREDUNC7; 2) sample the posterior parameter covariance matrix, and generate multiple calibration-constrained random parameter sets with RANDPAR; 3) recalibrate each parameter set with PEST up to achieving a level of fit fairly similar to the original calibration result (a tolerance of +5 % for the measurement objective function is allowed by MFIT); and 4) compute histograms of the recalibrated parameter values. The following two assumptions underlie this method: (i) the upper and lower parameter bounds specified by the user for the PEST inversion reflect the prior (expert knowledge) parameter uncertainty, and (ii) the model parameters are statistically independent from a prior point of view. This second assumption is relaxed through the recalibration process.

## 4 Code verification

The robustness of the PEST inversion program has been demonstrated in a number of studies, see, e.g., Anderson et al. (2015) and Hunt et al. (2019), and is not reassessed here. The purpose of this section is to assess the accuracy of MFIT direct simulations through five synthetic test cases. Tests 1 and 2 address the case of a single flow channel described as a single-porosity medium in which the transport is governed by advection-dispersion. An instantaneous injection of the tracer is assumed in test 1, whereas test 2 addresses the case of an exponentially decaying concentration at the inlet. A double-porosity medium, single flow channel is assumed in tests 3 and 4, which conform to the assumptions of the SFDM and 2RNE model, respectively. In test 5, the tracer is transported by advection-dispersion in a multiflow system composed of two channels. This scenario corresponds to the MDM. The input parameters for the five test cases are listed in Table 2. The BTCs simulated by MFIT for tests 1, 2 and 4 are compared to those obtained by CXTFIT. The MFIT simulations for tests 3 and 5 are compared against those obtained by TRACI. As shown in Fig. 2, very good agreement was obtained in each case.

**Table 2.** Input parameters for the five verification tests.

| Test | Parameters | Values |
|---|---|---|
| **1**<br>**(single flow channel, ADE,**<br>**instantaneous injection)** | Flow rate $Q$ | 10 m$^3$ h$^{-1}$ |
| | Injected mass $m$ | 20 g |
| | Mean transit time $T_0$ | 200 h |
| | Peclet number $Pe$ | 2 |
| **2**<br>**(single flow channel, ADE,**<br>**exponentially decaying injection)** | Mean transit time $T_0$ | 70 h |
| | Peclet number $Pe$ | 10 |
| | Initial (maximum) injection concentration $C_0$ | $8.0 \times 10^{-3}$ mg l$^{-1}$ |
| | Gamma coefficient $\gamma$ | 0.9 |
| **3**<br>**(single flow channel, SFDM)** | $Q$, $m$, $T_0$, $Pe$ | same as Test 1 |
| | Diffusion parameter $\beta$ | 0.04 h$^{-1/2}$ |
| **4**<br>**(single flow channel, 2RNE)** | $Q$, $m$, $T_0$, $Pe$ | same as Test 1 |
| | Length of the flow channel $L$ | 1000 m |
| | Fraction of mobile water $\psi$ | 0.7 |
| | Omega coefficient $\omega$ | 0.1 m$^{-1}$ |
| **5**<br>**(two channels, MDM-ADE)** | Total system flow rate $Q$ | 10 m$^3$ h$^{-1}$ |
| | Mass flowing through the first channel $m_1$ | 12 g |
| | Mass flowing through the second channel $m_2$ | 8 g |
| | Mean transit time in the first channel $T_{01}$ | 170 h |
| | Mean transit time in the second channel $T_{02}$ | 300 h |
| | Peclet number in the first channel $Pe_1$ | 15 |
| | Peclet number in the second channel $Pe_2$ | 80 |

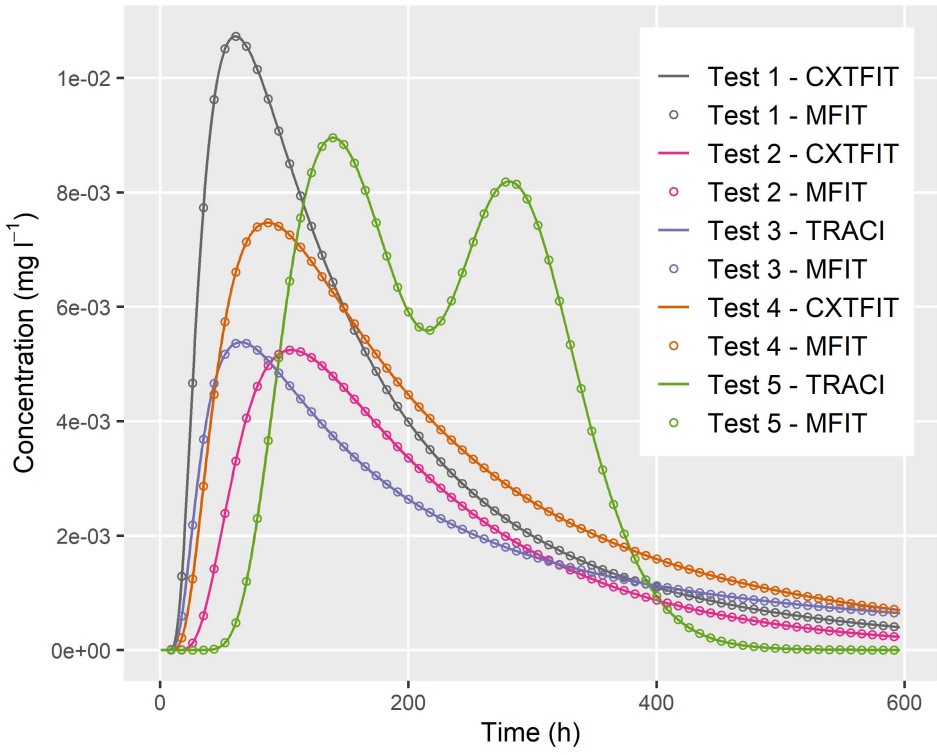

**Figure 2.** Comparison among MFIT, CXTFIT, and TRACI simulations for test 1 (single flow channel, ADE, instantaneous injection), test 2 (single flow channel, ADE, exponentially decaying injection), test 3 (single flow channel, SFDM), test 4 (single flow channel, 2RNE), and test 5 (two channels, MDM-ADE)

## 5 Assessment of the multistart optimization method

The purpose of this section is to assess the automatic multistart method described in section 3 using a new synthetic test case. A multimodal BTC that corresponds to 3 channels has been simulated using the MDMi program with the parameters listed in Table 3. A "blind" inversion of this BTC has been performed using the automatic multistart method with a maximum number of flow channels $N_{max} = 6$. The only model parameter that has been fixed prior to the inversion process was the total flow rate $Q$ to simplify the post-comparison of the inverted mass values in each channel with the "true" mass values. Otherwise, a degree

of freedom would persist for the pairs of the optimized $Q$ and $m_j$ values, i.e., multiplying or dividing these parameters by the same constant would yield the same BTCs; refer to Eq. (6). The parameters $m_j$, $T_{0j}$ and $Pe_j$ of the different flow channels were optimized with virtually no upper and lower bound constraints (minimum and maximum allowed parameter values of $1.0 \times 10^{-10}$ and $1.0 \times 10^{+10}$, respectively). As shown in Fig. 3, the inverted BTCs that correspond to $N = 3, 4, 5,$ and 6 channels overlap perfectly with each other and with the original simulated BTC; and as shown in Table 4, the optimized values for the

parameters of the 3-channel model are equal to the "true" parameter values.

**Table 3.** Model parameters that correspond to the multimodal simulated BTC in Fig. 3.

| Parameters | Values |
| --- | --- |
| $Q$ | $10 \ \text{m}^3\,\text{h}^{-1}$ |
| $m_1$ | 10 g |
| $m_2$ | 6 g |
| $m_3$ | 4 g |
| $T_{01}$ | 150 h |
| $T_{02}$ | 250 h |
| $T_{03}$ | 350 h |
| $Pe_1$ | 20 |
| $Pe_2$ | 50 |
| $Pe_3$ | 100 |

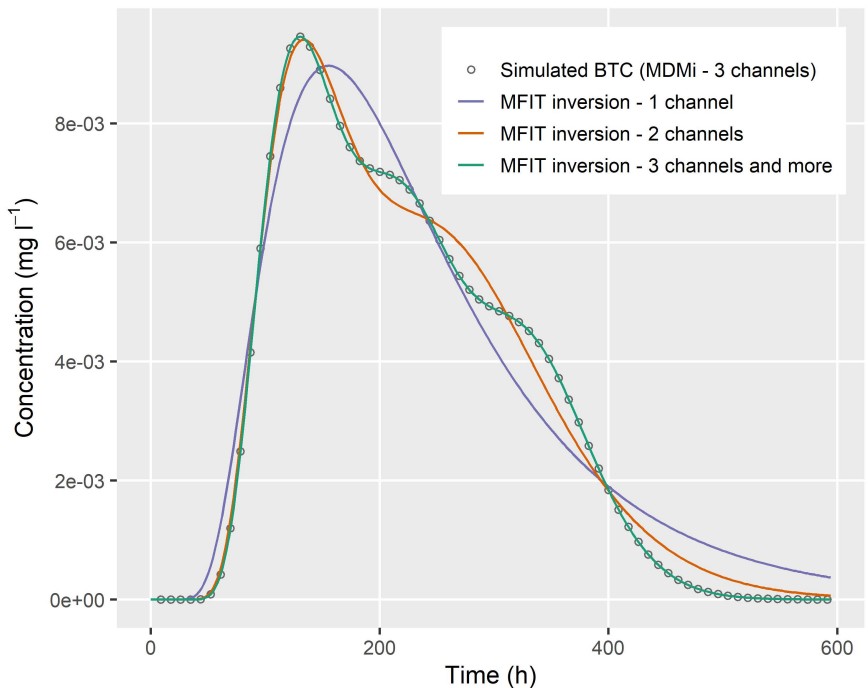

**Figure 3.** Inversion of the 3-channel-simulated BTC using the automatic multistart method with $N_{max} = 6$. The inverted BTCs that correspond to $N = 3, 4, 5,$ and 6 channels overlap perfectly with each other and the original simulated BTC.

**Table 4.** Optimized model parameters that correspond to the inverted BTCs in Fig. 3.

| $N$ | 1 | 2 | 3 | 4 | 5 | 6 |
|---|---|---|---|---|---|---|
| $m_1$ (g) | 21.11 | 10.79 | 10.00 | 2.79 | 2.66 | 2.66 |
| $m_2$ (g) | - | 9.54 | 6.00 | 7.19 | 7.35 | 7.35 |
| $m_3$ (g) | - | - | 4.00 | 6.02 | 5.99 | 5.91 |
| $m_4$ (g) | - | - | - | 4.00 | 2.58 | 2.62 |
| $m_5$ (g) | - | - | - | - | 1.42 | 1.45 |
| $m_6$ (g) | - | - | - | - | - | 0.02 |
| $T_{01}$ (h) | 239.36 | 155.82 | 150.00 | 126.17 | 151.55 | 151.31 |
| $T_{02}$ (h) | - | 302.91 | 250.00 | 158.91 | 149.47 | 149.60 |
| $T_{03}$ (h) | - | - | 350.00 | 250.00 | 249.97 | 249.30 |
| $T_{04}$ (h) | - | - | - | 350.01 | 349.48 | 347.68 |
| $T_{05}$ (h) | - | - | - | - | 350.84 | 351.08 |
| $T_{06}$ (h) | - | - | - | - | - | 405.58 |
| $Pe_1$ | 6.72 | 17.52 | 20.00 | 24.22 | 19.80 | 19.92 |
| $Pe_2$ | - | 27.55 | 50.00 | 22.18 | 20.07 | 20.01 |
| $Pe_3$ | - | - | 100.00 | 49.94 | 50.04 | 50.62 |
| $Pe_4$ | - | - | - | 100.00 | 98.45 | 88.42 |
| $Pe_5$ | - | - | - | - | 102.68 | 120.27 |
| $Pe_6$ | - | - | - | - | - | 442.13 |

## 6 Application example: analysis of tracer BTCs from the Hydrogeological Experimental Site in Poitiers, France

The HES is a field research facility operated by the University of Poitiers, France. The facility consists of 32 wells that have
been drilled within an overall area of 0.2 km² (Fig. 4) and fully penetrate a 100-m-thick confined limestone aquifer. The
interwell flow and transport connectivity have been shown to be mainly related to karst conduits, 0.01–3 m in diameter, that
develop preferentially within specific lithostratigraphic horizons interbedded with nonkarstified limestone units. The karstified
layers may contribute to the connectivity from one well to another either directly (e.g., the wells intersect with the same karst
network in a single layer) or indirectly (e.g., the wells intersect with different karst network layers that are interconnected by
either a third well or a subvertical fracture); see Audouin et al. (2008) and Chatelier et al. (2011).

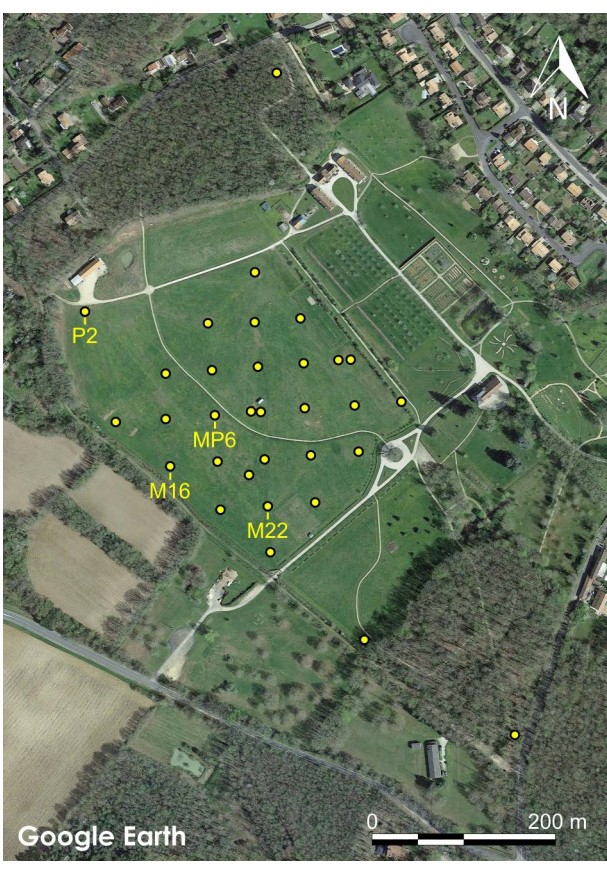

**Figure 4.** Locations of wells at the HES in Poitiers, France. Map data are from Google.

A large number of pumping test experiments have been conducted at the HES since 2002. As discussed in a number of studies (Delay et al., 2007; Riva et al., 2009; Delay et al., 2011; Bodin et al., 2012; Sanchez-Vila et al., 2016; Le Coz et al., 2017), the drawdown responses exhibit complex behaviors, which are likely due to the strong aquifer heterogeneity induced predominantly by the presence of karst features. In addition to the pumping test experiments, a number of cross-borehole tracer tests have been performed at the HES since 2011. The standard experimental protocol of HES tracer experiments can be summarized as follows:

1. Starting a pumping experiment and waiting for the establishment of a pseudosteady state flow regime (i.e., stabilization of interwell piezometric head gradients), which typically takes approximately 6 h at the HES;

2. Performing flow log measurements in the candidate injection well to identify the main inflow/outflow levels along the well bore;

3. Connecting a series of 2.5 m length and 1.5 cm inner diameter PVC pipes in the injection well, from the ground down to the tracer injection depth (usually chosen to be as close as possible to a main outflow level). The pipeline is terminated by a 5 cm length screened cap that ensures a horizontal outflow of the tracer solution in the injection well.

4. Injecting a tracer solution (typically 2 l of uranine solution at 1 g l$^{-1}$) in the pipe and flushing with 40 l of "clean" ground water. The total duration of an injection is typically less than 3 min.

5. Monitoring the tracer BTC at the pumped well using a flow-through fluorometer (Albillia GGUN-FL22) connected to a branch pipe extending from the discharge line at ground level. The fluorometer is periodically calibrated in the laboratory with solutions of 10 µg l$^{-1}$ and 100 µg l$^{-1}$.

To date, more than 70 cross-well tracer experiments have been performed at the HES. The purpose here is not to interpret each of these experiments but to pick a few examples for illustrating the application of the MFIT software. The selected data correspond to three tracer experiments that were performed in 2016 and 2017 using well M22 as the pumped well and M16, MP6, and P2 as injection wells. Fig. 5 shows the experimental BTCs and a collection of calibrated MFIT curves for different numbers of channels. The selected experiments were chosen for their representativeness of the BTC shapes observed at the HES, which exhibit either a single peak followed by a more or less pronounced tailing, e.g., P2-M22, overlapping double-peak responses, e.g., M16-M22, or well-marked multimodal responses, e.g., MP6-M22. The mass recovery ratios for these three tracer experiments were 58 %, 79 %, and 60 %, respectively. Note that these recovery data cannot be included in the model because the flow structure assumption that underlies the multiflow approach (Fig. 1) implies that all the mass that enters the system flows out after a certain lapse of time. The same holds for any single- or double-porosity modeling approach based on a 1-D flow assumption. For tracer tests that are performed in steady state conditions and involve non-reactive tracers, an incomplete recovery of the injected mass indicates a diverging flow structure between the injection site and the monitoring point. Unfortunately, no additional information can be obtained about this flow divergence from the tracer data only. Therefore, the total mass in a multiflow model must be consistent with the recovered tracer mass rather than the injected mass.

The model fit results shown in Fig. 5 were obtained using the multistart method discussed in section 3 and only SVD as a regularization tool for the inversion. None of the model parameters were fixed, and all were optimized within realistic upper and lower limits. The optimized parameter values and their composite sensitivities at the end of the optimization process are provided in the Supplement (Table S1). Unsurprisingly, the model parameters that influence the spreading of transit/residence times in the individual flow channels while accounting for different processes ($Pe$, $\gamma$, $\beta$, $\psi$, and $\omega$) are sensitive to the number of channels. For instance, when comparing single- with multiple-channel models, the former requires lower $Pe$ values to compensate for the coarser description of the flow system heterogeneity (recall that the dispersion coefficient integrated in the Peclet number reflects the unresolved variability of the flow velocity below the modeling scale). The same observation holds when comparing single- and double-porosity models with the same number of flow channels, i.e., the $Pe$ values of single-porosity models are lower than the $Pe$ values of double-porosity models because part of the spreading of transit/residence times in the latter case is implicitly captured by solute mass exchanges between the mobile and immobile domains. A noticeable exception is the diffusion parameter $\beta$ of the SFDM model, whose values are mostly around $1.0 \times 10^{-3}$ h$^{-1/2}$. This value corresponds to the upper bound of the optimization range set for this parameter, which is based on a matrix porosity of 30 %, a molecular diffusion coefficient of $1.0 \times 10^{-9}$ m$^2$ s$^{-1}$, and a flow-channel aperture of $1.0 \times 10^{-2}$ m. Beta values larger than $1.0 \times 10^{-3}$ h$^{-1/2}$ would be physically unrealistic. The fact that the Beta value is limited by its upper bound during the optimization

process indicates that the SFDM model is not suitable for describing the HES tracer experiments, as further discussed below. All other parameters have converged to values far from their optimization bounds.

Beyond what can be visually inferred from Fig. 5, the assessment of the relative fitting performance of the different models can be analyzed through the evolution of the measurement objective function, hereafter named PHI, with respect to the number $N$ of channels and/or the number $P$ of optimized model parameters. Fig. 6 displays the PHI($N$) and PHI($P$) curves summarizing

the best-fitting results achieved with the multistart PEST optimization and the SCEUA_P and CMAES_P global optimization routines. A number of observations can be made from this figure. As a first remark, the SCEUA curves for the two MDP models are missing in Fig. 6. The reason is that the SCEUA_P program has no "forgive−error" capability, i.e., if a set of trial parameters causes the numerical evaluation of Eq. (11) or Eq. (15) to crash, the optimization process is stopped instead of moving to a new set of parameter values. Such a forgiveness option is available in the PEST and CMAES_P programs. The

next observations that can be made from Fig. 6 are that the CMAES and SCEUA curves are (i) more irregular, (ii) always above or equal to their PEST-computed counterparts, and (iii) do not always follow the expected decreasing trend in the PHI value (meaning a better model fit) as the number of channels rises, as depicted by the PEST curves. However, it must be mentioned that the number of optimization runs was much greater for the CMAES_P and SCEUA_P programs, and various optimization options were tested (e.g., changing the upper and lower parameter bounds and log-transformation of parameters).

The CMAES and SCEUA curves shown in Fig. 6 are actually the "best results" obtained after several days of computation time. It is clear that the multistart PEST optimization method performs better in each case.

The PHI curves obtained by PEST can be viewed as Pareto curves illustrating the tradeoff between the model fitting quality and the number of channels or the number of calibration parameters. It must be noted that since no Tikhonov regularization was used in this illustration example, the model inversion results for higher $N$ values are likely affected by overfitting. More

reliable parameter values could be obtained by adding Tikhonov regularization constraints to the optimization process.

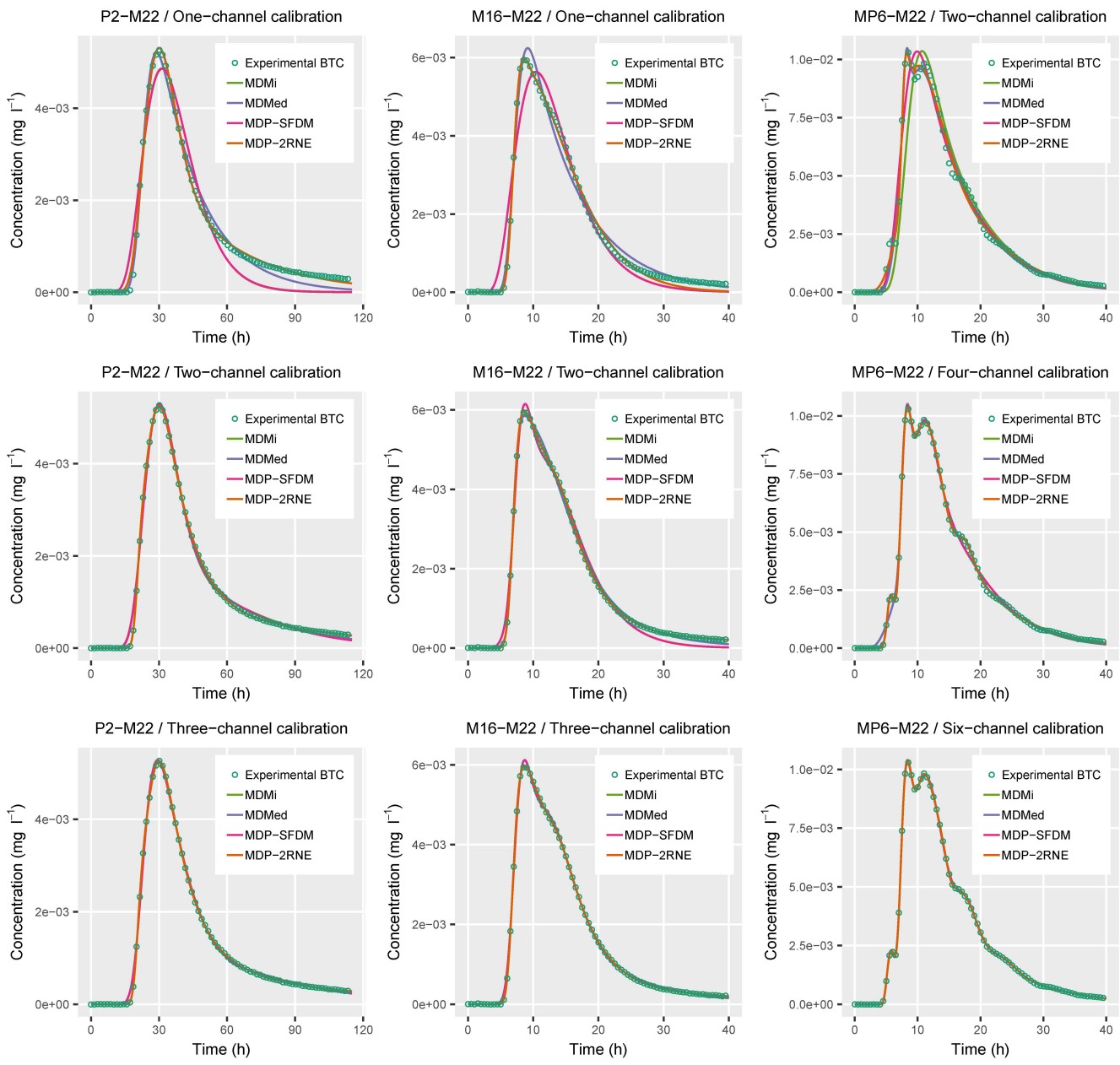

**Figure 5.** Inversion solutions of three tracer BTCs for different numbers of channels. Some model curves are hardly distinguishable, as they perfectly overlap (refer to the text and Fig. 6).

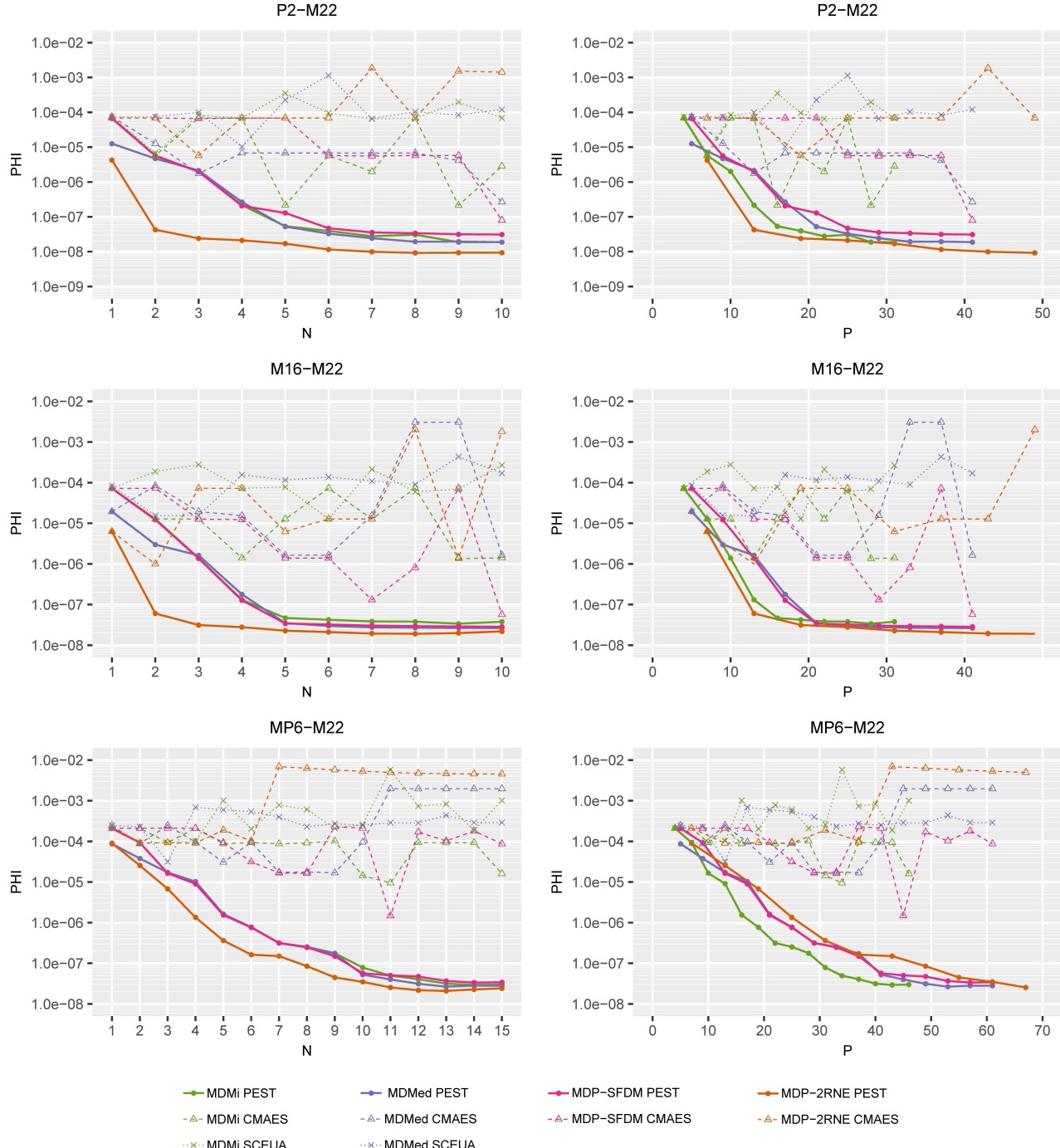

**Figure 6.** Best-fitting performance of the multiflow models achieved using PEST with the multistart optimization approach and using global optimizers. $N$ is the number of channels in the models, $P$ is the number of optimized parameters, and PHI is the sum of the squared weighted differences between the tracer BTCs and the model-fitted curves.

According to the PHI(*N*) curves shown in Fig. 6, the MDMi model and MDP-SFDM perform similarly for the three tracer tests, and the related PHI(*N*) curves are hardly differentiable. This result was expected, as the short duration of the HES tracer

tests, typically from a few hours to a few days, makes the matrix diffusion process unlikely to be significant. Assuming exponential decaying (MDMed) instead of instantaneous (MDMi) injection gives slightly better fitting results for a low number of channels but provides no benefit for a moderate to high number of channels. According to the PHI(*N*) curves, the fitting performance of the MDP-2RNE model seems significantly better than that of the three other models. However, this observation must be counterbalanced by the larger number of calibration parameters in the MDP-2RNE model (see Table 1). A two-channel

MDP-2RNE model involves 13 parameters, which corresponds to the number of parameters in a four-channel MDMi model. The PHI(*P*) curves shown in Fig. 6 provide a fairer assessment of the fitting performance of the different models. According to these curves, the MDP-2RNE model performs slightly better than the MDMi model for the P2-M22 tracer test (single-peak slightly tailed BTC), almost equally well for the M16-M22 tracer test (overlapping double peaks), and worse for the MP6-M22 tracer test (well-marked multimodal BTC). It must be appreciated that these two models should not be opposed to each

other. Both models likely provide an equally valid description of the tracer transport in the HES aquifer while relying on different conceptualizations of the medium heterogeneity.

The Pareto curves in Fig. 6 indicate that the *final choice* of a model, if one is to be made, relies on a tradeoff between the desired fitting accuracy and the desired degree of simplification/complexity with respect to the model structure (number of channels and/or number of model parameters). Beyond this subjective (expert) decision, which may depend on the goal of the

study, and therefore, will not be discussed further in the present application case, uncertainty remains in the inverted model parameters as a consequence of the nonuniqueness of the inverse problem. This uncertainty is related to both the equifinality of the model parameters, which is partly due to the multiflow framework structure, and the measurement noise in the tracer BTCs. Figures 7 and 8 illustrate the post-calibration uncertainty analysis capabilities of MFIT, via an assessment of the MDMi and MDP-2RNE model fittings of the M16-M22 tracer BTC with 1, 2, and 3 flow channels. Owing to the balance between the

$Q$ and $m_j$ terms in the model equation (Eqs. (6) and (15)), at least one of these parameters must be fixed to assess the uncertainty of the other parameters. Here, the value of $Q$ was set to 25 m$^3$ h$^{-1}$, which ensures the consistency of the model against the recovered tracer mass that was independently calculated from the experimental data (refer to Table S1). Following the PEST optimization of the different model parameters, 500 calibration-constrained parameter fields were stochastically generated and recalibrated by PEST. Depending on the model (MDMi or MDP-2RNE) and number of flow channels, between 483 and 500

recalibration runs successfully achieved a level of fit that is fairly similar (i.e., within a tolerance of +5 % for the PHI value; refer to section 3) to that associated with the original calibration parameter field. The histograms shown in Figs. 7 and 8 were constructed from these recalibrated parameter sets and illustrate the multitude of parameter combinations that are equally good, for a given number of flow channels, in terms of fitting the M16-M22 tracer BTC. As shown in these figures, the confidence intervals are quite narrow for most parameters but tend to widen as the number of channels increases, which reflects the

equifinality of the multiflow modeling approach. Although not shown here, it has been established that the tailed behaviors of the parameters $L_j$ and $\omega_j$ in Fig. 8 are due to a partial correlation between these two parameters (refer to Eq. (15)), i.e., fixing

the value of one parameter prior to the inversion drastically reduces the uncertainty of the other parameter. As previously discussed, the higher Pe values in Fig. 8 compared to Fig. 7 are due to the fact that the distribution of the transit/residence times with the 2RNE model is primarily controlled by the solute mass exchanges between the mobile and immobile domains.

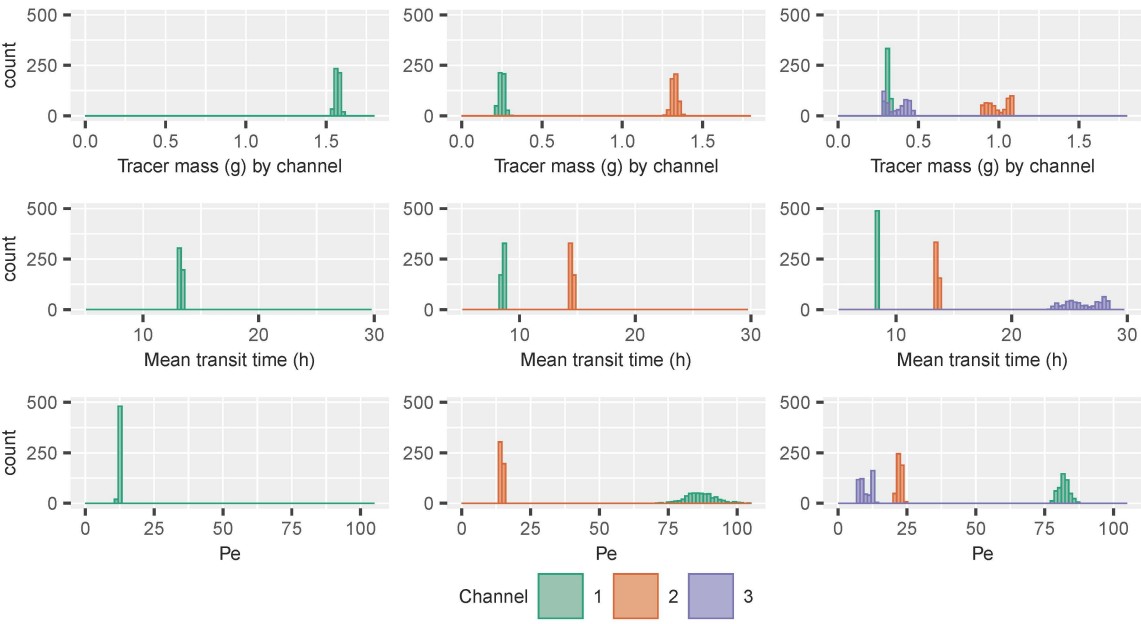


**Figure 7.** Postcalibration uncertainty of model parameter values for the inversion of the M16-M22 tracer BTC by the MDMi model with 1, 2, and 3 flow channels.

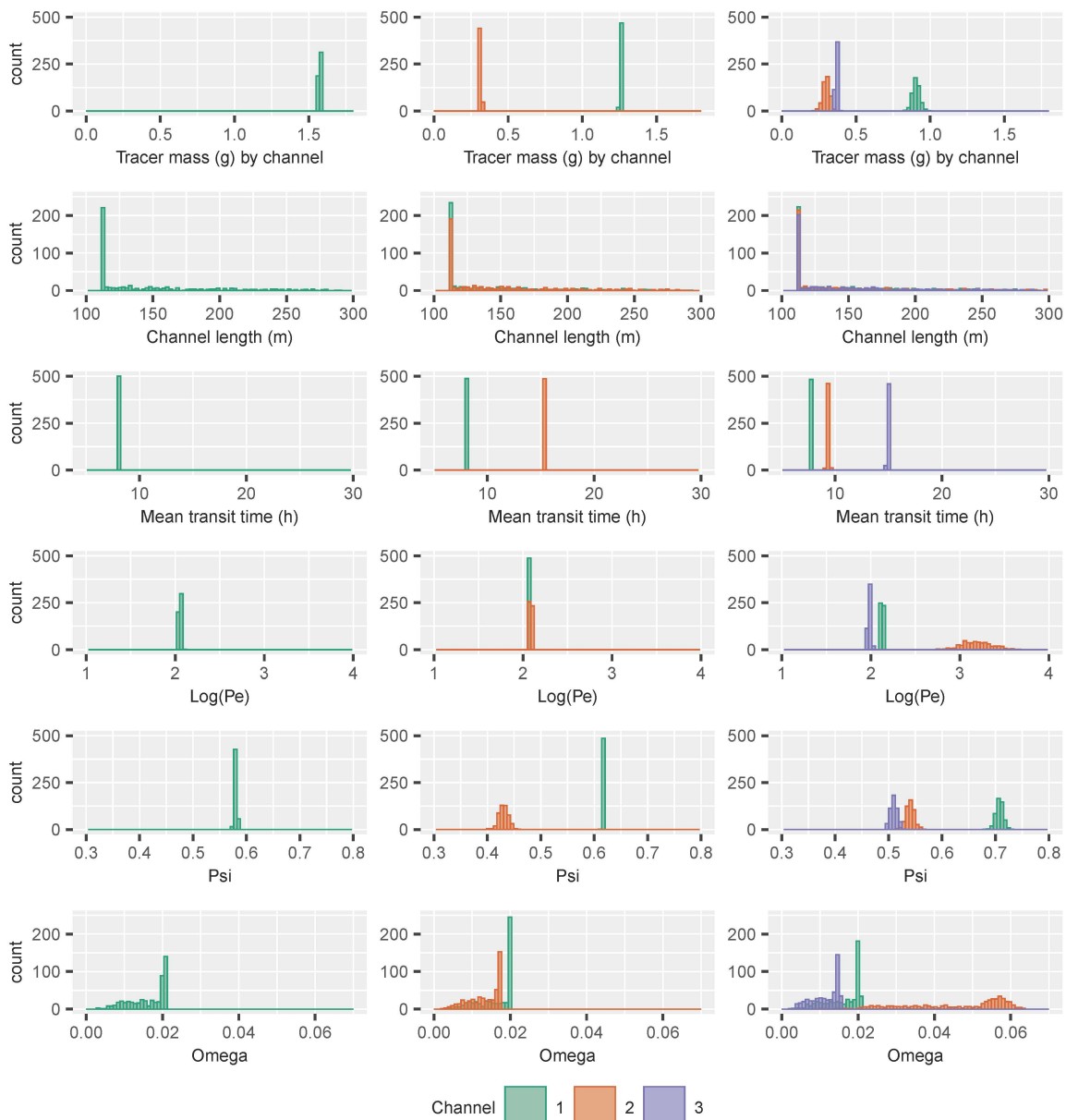

**Figure 8.** Postcalibration uncertainty of model parameter values for the inversion of the M16-M22 tracer BTC by the MDP-2RNE model with 1, 2, and 3 flow channels. A logarithmic scale has been employed for Pe due to a wider range of values than shown in Fig. 7.

# 7 Summary and conclusions

Multiple flow path transport is likely the rule rather than the exception in most transport problems in fractured and karst aquifers. The main aim of this paper was to present a new curve-fitting tool for the analytical modeling of BTCs from tracer tests performed in such media. The MFIT software is a free open-source Windows-based GUI that provides access to four multiflow transport models. The multiflow approach assumes that the transport from the injection site to the monitoring point takes place in a number of independent 1-D channels. The channels are not assumed to represent individual fractures or karst conduits but are lumped submodels of the main flow routes used by the tracer through the fractures/karst conduit network. The multiflow modeling framework allows the simulation of multimodal BTCs, which are frequently observed in fractured and karst aquifers. Two of the MFIT transport models combine the multiflow framework and the double-porosity concept, which is applied at the scale of the individual channels. This modeling approach, which has been named MDP, is believed to be new and versatile for the fitting of BTCs with multiple local peaks and/or extensive backward tailing. The accuracy of the MFIT-computed BTCs was verified against two other well-accepted simulation tools for five synthetic test cases.

An important feature of MFIT is its compatibility and interface with the advanced calibration tools of the PEST suite of programs. Hence, MFIT is the first BTC fitting tool that allows regularized inversion and nonlinear analysis of the postcalibration uncertainty of model parameters. Given the nonlinearity of the MFIT model equations, an original multistart algorithm was implemented to maximize the chances for PEST to converge to the global optimal solution in the parameter space during a BTC fitting procedure. The main drawback of the multistart optimization method is that the processing time can be long (up to a few hours) if a large number of channels is assumed in the model. Time reduction for this method is one of the development perspectives of the MFIT code, as the multistart process is computationally parallelizable. Other development perspectives are the management of more complex injection signals, e.g., described as multiple steps, and the implementation of additional analytical transport models for the simulation of reactive transport processes.

Three tracer test BTCs from the HES in Poitiers, France, were used for illustrating the application of the MFIT software. An analysis of the Pareto curves between the model fitting quality and the number of model calibration parameters suggests that the MDMi and MDP-2RNE models are the most appropriate for the interpretation of HES tracer tests. This preliminary result needs to be refined or confirmed by the analysis of additional HES tracer BTCs.

## Appendix A: Glossary

**Table A1.** Acronyms and model abbreviations utilized in the text.

| Acronym or model name | Description | Reference |
|---|---|---|
| ADE | Advection-dispersion equation | Zheng and Bennett (2002) |
| BTC | Breakthrough curve | |

| | | |
|---|---|---|
| CATTI | Computer Aided Tracer Test Interpretation: a computer program for tracer BTC fitting | Sauty et al. (1992) |
| CMA-ES | Covariance Matrix Adaptation – Evolution Strategy: a global optimization algorithm | Hansen and Ostermeier (2001) |
| CMAES_P | PEST-compatible program that implements the CMA-ES method | Doherty (2019a) |
| CXTFIT | Computer program for tracer BTC fitting | Toride et al. (1999) |
| DADE | Dual-advection-dispersion equation | Field and Leij (2012) |
| FEFLOW | Finite Element FLOW model; a simulation package for flow, heat, and mass transport in groundwater | Diersch (2014) |
| GUI | Graphical user interface | |
| HES | Hydrogeological Experimental Site in Poitiers, France | Audouin et al. (2008) |
| MDM | Multi-Dispersion Model | Maloszewski et al. (1992) |
| MDMed | Computer program that implements the Multi-Dispersion Model and assumes a non-instantaneous injection (exponentially decaying concentration) at the inlet of the flow system | This article |
| MDMi | Computer program that implements the Multi-Dispersion Model and assumes an instantaneous injection of tracer at the inlet of the flow system | This article |
| MDP | Multi-Double Porosity: a combination of multiflow and double-porosity models | This article |
| MDP_SFDM | Computer program that implements the MDP approach, where the mass exchanges between the mobile and immobile domains are modeled as a second-order (diffusion) process | This article |
| MDP_2RNE | Computer program that implements the MDP approach, where the mass exchanges between the mobile and immobile domains are modeled as a first-order process | This article |
| MFIT | MultiFlow Inversion of Tracer breakthrough curves: a GUI for the MDMi, MDMed, MDP_SFDM, MDP_2RNE, and PEST programs. | This article |
| MIM | Mobile-Immobile Model | Coats and Smith (1964) |
| MODFLOW | MODular three-dimensional groundwater FLOW model: a computer code developed by the U.S. Geological Survey that numerically solves the groundwater flow equation | Langevin et al. (2017) |
| MT3DMS | Modular Three-Dimensional MultiSpecies transport model: a numerical code to simulate solute transport in groundwater | Zheng et al. (2012) |
| OM-MADE | One-dimensional Model for Multiple Advection, Dispersion, and storage in Exchanging zones: a python script to simulate solute transport in multiflow systems with possible mass exchanges between the flow channels | Tinet et al. (2019) |
| OptSFDM | Computer program for tracer BTC fitting based on the SFDM model | Gharasoo et al. (2019) |
| OTIS | One-dimensional Transport with Inflow and Storage: a numerical code to simulate solute transport in streams and rivers | Runkel (1998) |

| PEST | Parameter ESTimation: a collection of computer programs for model-independent parameter estimation and uncertainty analysis | Doherty (2019a) |
| SCE-UA | Shuffled Complex Evolution method – University of Arizona: a global optimization algorithm | Duan et al. (1992) |
| SCEUA_P | PEST compatible program that implements the SCE-UA method | Doherty (2019a) |
| SFDM | Single-Fracture Dispersion Model | Maloszewski and Zuber (1990) |
| STANMOD | STudio of ANalytical MODels: a collection of computer programs for tracer BTC fitting | van Genuchten et al. (2012) |
| SVD | Singular value decomposition | Doherty (2015) |
| TRAC | Computer program for tracer BTC fitting | Gutierrez et al. (2013) |
| TRACI | Computer program for tracer BTC fitting | Käss (2004) |
| 1-D | One-dimensional | |
| 2RNE | Two-region nonequilibrium equation | Toride et al. (1993) |

**Table A2.** List of model parameters.

| Parameter | Description | Unit | Specific model (an empty box means that the parameter is employed in all the models) |
|---|---|---|---|
| $b_j$ | Half-aperture of the $j$-th flow channel | L | MDP-SFDM |
| $C_j$ | Concentration in the $j$-th flow channel | $ML^{-3}$ | |
| $C_{pj}$ | Concentration in the immobile domain assigned to the $j$-th channel | $ML^{-3}$ | MDP-SFDM |
| $C_{imj}$ | Concentration in the immobile domain assigned to the $j$-th channel | $ML^{-3}$ | MDP-2RNE |
| $C_0$ | Initial (maximum) concentration at the inflow boundary for an exponentially decaying injection concentration | $ML^{-3}$ | MDMed |
| $D_j$ | Dispersion coefficient in the $j$-th flow channel | $L^2T^{-1}$ | |
| $D_{pj}$ | Molecular diffusion coefficient in the immobile domain assigned to the $j$-th channel | $L^2T^{-1}$ | MDP-SFDM |
| $L_j$ | Length of the $j$-th flow channel | L | |
| $m_j$ | Part of the solute mass flowing through the $j$-th channel | M | MDMi, MDP-SFDM, MDP-2RNE |
| $N$ | Number of flow channels | - | |
| $N_{max}$ | Maximum number of flow channels | - | |
| $Pe_j$ | Peclet number in the $j$-th channel | - | |
| $P$ | Number of optimized model parameters | - | |
| PHI | Measurement objective function (sum of the squared weighted differences between the tracer BTCs and the model-fitted curves) | $M^2L^{-6}$ | |
| $Q$ | Total system flow rate | $L^3T^{-1}$ | |
| $Q_j$ | Flow rate in the j-th channel | $L^3T^{-1}$ | |
| $t$ | Time variable | T | |

| | | | |
|---|---|---|---|
| $T_{min}$ | Minimum time value of the user-provided BTC | T | |
| $T_{max}$ | Maximum time value of the user-provided BTC | T | |
| $T_5$ | $T_5$ time, Eq. (18) | T | |
| $T_{5th}$ | Earliest time at which the concentration values exceed 5 % of the maximum concentration value | T | |
| $T_{95}$ | $T_{95}$ time, Eq. (19) | T | |
| $T_{95th}$ | Latest time at which the concentration values exceed 5 % of the maximum concentration value | T | |
| $T_{0j}$ | Mean transit time in the $j$-th channel | T | |
| $u_j$ | Advection velocity in the $j$-th flow channel | $LT^{-1}$ | |
| $x_j$ | Spatial coordinate along the $j$-th flow channel | L | |
| $y_j$ | Spatial coordinate perpendicular to the $j$-th flow channel | L | |
| $\alpha_j$ | First-order mass transfer coefficient between the mobile and immobile domains assigned to the $j$-th channel | $T^{-1}$ | MDP-2RNE |
| $\beta_j$ | Diffusion parameter in the $j$-th flow channel, Eq. (12) | $T^{-1/2}$ | MDP-SFDM |
| $\gamma_j$ | Gamma coefficient in the $j$-th flow channel, Eq. (8) | - | MDMed |
| $\theta_j$ | Volumetric water content of the mobile domain assigned to the $j$-th channel | - | MDP-2RNE |
| $\theta_{imj}$ | Volumetric water content of the immobile domain assigned to the $j$-th channel | - | MDP-2RNE |
| $\lambda_j$ | Time decay constant that controls the exponentially decaying release of tracer in the $j$-th channel | $T^{-1}$ | MDMed |
| $\xi$ | Integration variable, Eq. (11) | T | MDP-SFDM |
| $\sigma_j$ | Standard deviation of travel times for transport by advection and dispersion in the $j$-th channel | T | |
| $\tau$ | Integration variable, Eq. (15) | L | MDP-2RNE |
| $\psi_j$ | Fraction of mobile water in the $j$-th channel, Eq. (16) | - | MDP-2RNE |
| $\omega_j$ | Omega coefficient in the $j$-th flow channel, Eq. (17) | $L^{-1}$ | MDP-2RNE |

**Code and data availability.** The source codes of the MFIT program suite version 1.0.0 are available from https://doi.org/10.5281/zenodo.3470751 under the terms of the CeCILL Free Software License Agreement v2.1 (https://spdx.org/licenses/CeCILL-2.1.html#licenseText, last accessed: 02 October 2019). An "EXE" installation package compiled with Inno Setup (http://www.jrsoftware.org/isinfo.php, last accessed: 02 October 2019) and a user's guide are provided along with the source codes. The following numerical libraries are required for the compilation of the MFIT suite of

codes: Boost (https://www.boost.org/, last accessed: 02 October 2019), GSL-GNU (https://www.gnu.org/software/gsl/, last accessed: 02 October 2019), and Spline (https://github.com/ttk592/spline, last accessed: 02 October 2019). The PEST program package is also required for running MFIT. PEST is distributed by default using the MFIT software installer or can be independently downloaded from http://www.pesthomepage.org/Downloads.php (last accessed: 02 October 2019). The data of

the HES tracer experiments processed in section 6 of this study are available from the H+ database
(http://hplus.ore.fr/en/poitiers/data-poitiers, last accessed: 02 October 2019) with registration of a free account.

**The Supplement related to this article is available online at** https://doi.org/10.5281/zenodo.3824439**.**

**Acknowledgments.** The HES field research facility is managed by Gilles Porel. The experimental protocol used for the HES tracer experiments was developed by Benoit Nauleau and Gilles Porel. The assistance of Benoit Nauleau, Gilles Porel, and Denis Paquet in conducting the tracer experiments is gratefully acknowledged. The author would like to thank the two
anonymous reviewers for their valuable comments and feedback, which helped improving the manuscript.

**Financial support.** This research was supported by the French National Observatory H+, the European Union (ERDF), and "Région Nouvelle Aquitaine".

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
