# Peer review of "MFIT 1.0.0: Multiflow inversion of tracer breakthrough curves in fractured and karst aquifers"

_Geoscientific Model Development, 2019_

## Referee Comment (RC1) · Anonymous Referee #1 · 20 Apr 2020

**GEOSCIENCES MODEL DEVELOPMENT**

**GMD-2019-286: MFIT 1.0.0: Multiflow inversion of tracer breakthrough curves in fractured and karst aquifers**

Firstly, thank you for the opportunity to review this interesting manuscript matching the field of tracing hydrology. Jacques Bodin presents a new software for BTC´s fitting of artificial tracer tests in karst and fractured aquifers. The software compiles four transport models able to advance in the simulation of single/multiple and long-tailed curve shapes. Individual models proceed from the modification of previous analytical and numerical solutions and two of them are novel since they couple the multiflow approach (several 1-D independent channels) with the double-porosity concept. Additionally, an advanced optimization interface using PEST tools is also included in the modelling package.

From my opinion, advances in the field of artificial tracing tests must be oriented to gain new insights about solute transport dynamics through highly heterogeneous media from the test design with new injection-sampling strategies, but also from the more precise interpretation of karst conduit system geometry. Further efforts are expected to better explain BTCs resulting from hydrodynamic processes other than advection and dispersion, considering a clear focus on physical processes rather than achieving a suitable mathematical/numerical model framework. Therefore, the new development of novel software for BTC´s fitting is considered a notable advance in the karst community, as it is the case of the present manuscript. The huge computation works performed by the author are noticeable to achieve transport analytical/numerical solutions to physically reproduce multi-peaks and long-tailed curves. Overall, the integration with an optimization module is increasingly demanded in such type of model approach to avoid trial-error direct simulations and very often the consequent lack of accurate results.

However, my major criticism is focused on the code verification, in particular in the BTCs selected for the comparative analysis of simulation results. The five synthetic BTCs generated fail in both the relatively simple curve morphologies and the test duration. Since the four proposed models try to better fit multi-peaks and long-tailed curve shapes, the multimodal curves obtained from real field experiences show more marked/pronounced peaks (very often reaching relatively quite similar tracer concentration, as twin peaks) and the long-tailed ones (even with higher concentrations slowly decreasing along the lower slope ending curve segments) use to be recordered during much more prolonged tests (>100 hours). So, I would recommend incorporates and/or replacing new synthetic BTCs representing more adapted-to-reality morphologies. This will deeply test the code efficiency under more realistic and non-ideal (Fickian) transport dynamics. Regarding the modeled BTCs from the HES experimental site, they also display short tracer test duration and local transport dynamics. Some questions arise me, what about longer –multi-kilometers- karst connections and their expected very often long-tailed BTCs? and, what about the degree of flow diversion in anastomosed/ forked karst conduit systems and their associated multi-peak BTCs? I agree with the proposed pathway decomposition in multi-single channel scheme but, how the flow diversion in one or several of them and where (close to the injection point or to the end of the master conduit) may condition the obtained BTC shape?

Moreover, I miss complementary numerical results such as transport parameters and their discussion (i.e. sensitivity analysis) for a deeper comparative analysis of simulation results in section 5. The recovery rate of the injected tracer for the three examined BTCs would be

helpful to the reader to have information about how many tracer mass has been lost during the test. This will help to understand the potential role of rock matrix or stagnant zones in the karst circuit by which anomalous transport is reflected as multi-peaks or long-tailed BTC shapes.

In terms of format, I have to say that the manuscript is generally well structured and balanced (regarding its principal sections), as well as correctly written in English language and no substantial grammatical deficiencies has been observed throughout the manuscript. Besides, I recommend adding at the early sections a glossary of acronyms and parameters described throughout the text.

In summary, I consider that the paper in its present form is suitable for its publishing in GEOSCIENCES MODEL DEVELOPMENT journal only if suggested recommendations incorporate to this version of the manuscript.

**Point-to-point comments:**

Page 12: Table 2, test 4 >>> "Partitioning coefficient (ß)" instead of "Fraction of mobile water (ψ)"?

Page 12: Table 2, test 4 >>> "Mass transfer coefficient" instead of "Omega coefficient"?

---

## Referee Comment (RC2) · Anonymous Referee #2 · 28 Apr 2020

This manuscript presents new software for modelling tracker data from fractured karst aquifers. I found it to be very well written and particularly well organised in the introduction and methods sections. Some minor improvements are needed to the figures, discussion of uncertainty results and potentially the code verification section. I have outlined these as part of some specific comments below:

The abstract and introduction is very clear, and the contribution of the paper carefully set out.

Figure 1: should this say what the dashed line represents, is it non flowing water? Line 115-116: "A possible reason is the increasing number of fitting parameters, which makes the inverse problem more complicated. The use of modern inversion tools such as PEST enables overcoming this problem, as discussed in section 3" I agree that

these methods can efficiently find parameters sets in the situation you outline but I would assume not without the possibility that the parameters best fitting the data are far from unique and more so the greater the number of parameters. For me, this sentence misses a discussion of this important caveat in an otherwise very carefully considered section.

Around line 235: For my understanding, is the optimisation run for a given number of channels and if so should the user seek the minimum number of n that perform well for the measurement objective function and regularisation terms. OK, I see later where this comes in but I'll leave the comment so you can see the issue I had when reading for the first time.

Around line 285: A series of utility functions are called here for the uncertainty analysis. I don't think they need further explanation here but a pointer to the relevant documentation/literature on these would aid completeness.

Section 4 code verification – should this also test for the case where n channels is unknown? So for test 5 if n_max was set to 6 would the same results be found as for the current test 6. Perhaps this goes beyond verification of the transport processes models, which is clearly the aim of this section, but I think checking the multistart would add value if feasible.

Figure 3: In my version the dots and labels overlap, generally this figure could be cleaner, and the scale bar is also quite small. Would it also be possible to highlight the wells used for pumping and injections in the experiments, perhaps with colours or different symbology.

Line 345: Do the parameter bounds come into play in the optimised parameter sets? i.e. do you get parameters optimising to the bounds? Generally, there is not any discussion of the parameters found, we there a reason for this? I think this should be justified.

[Figure]

Figure 5: Could the legend be a single legend for all plots. On my version the legend is also quite small making dashed and continuous lines difficult to identify. Worth checking in the final production of the figure for publication.

Line 390 uncertainty analysis – Could you be more explicit about why the particular model and test case was chosen for the uncertainty analysis.

Line 396: "fairly similar" could you be more precise about how similar was defined. The uncertainty analysis description in the methods is quite brief which means its difficult to fully appreciate the setup here in my opinion.

Around line 400 – I feel the discussion of these results is somewhat rushed regarding the uncertainty analysis, I don't feel I fully appreciate the results. Is this conclusion made because only the four and six channel models capture the first peak? MDP-2RNE seems to for the two channels although it's difficult to see if this is really the case.

---

## Author Comment (AC1) · 24 May 2020

**Author's response to reviews of the manuscript "MFIT 1.0.0: Multiflow inversion of tracer breakthrough curves in fractured and karst aquifers" by Jacques Bodin**

jacques.bodin@univ-poitiers.fr

I am grateful to the two referees for their positive and constructive feedback, which has greatly benefited the work presented in this manuscript. I address their specific comments in detail here.

Font legend:
1. *Reviewers' comments are shown in a black italic font.*
2. My responses are shown in a blue normal font. Where not specified, line, table and/or figure numbers refer to the original manuscript.
3. The text that has been modified or added to the revised manuscript is shown in an orange normal font.

**Anonymous Referee #1**

*My major criticism is focused on the code verification, in particular in the BTCs selected for the comparative analysis of simulation results. The five synthetic BTCs generated fail in both the relatively simple curve morphologies and the test duration. Since the four proposed models try to better fit multi-peaks and long-tailed curve shapes, the multimodal curves obtained from real field experiences show more marked/pronounced peaks (very often reaching relatively quite similar tracer concentration, as twin peaks) and the long-tailed ones (even with higher concentrations slowly decreasing along the lower slope ending curve segments) use to be recordered during much more prolonged tests (>100 hours). So, I would recommend incorporates and/or replacing new synthetic BTCs representing more adapted-to-reality morphologies. This will deeply test the code efficiency under more realistic and non-ideal (Fickian) transport dynamics.*

The model parameter values in tests 1−5 have been modified following your recommendations. Table 2 and Fig. 2 have been updated.

**Table 2.** Input parameters for the five verification tests.

| Test | Parameters | Values |
|---|---|---|
| **1**
 **(single flow channel, ADE, instantaneous injection)** | Flow rate $Q$ | 10 m$^3$ h$^{-1}$ |
| | Injected mass $m$ | 20 g |
| | Mean transit time $T_0$ | 200 h |
| | Peclet number $Pe$ | 2 |
| **2**
 **(single flow channel, ADE, exponentially decaying injection)** | Mean transit time $T_0$ | 70 h |
| | Peclet number $Pe$ | 10 |
| | Initial (maximum) injection concentration $C_0$ | $8.0 \times 10^{-3}$ mg l$^{-1}$ |
| | Gamma coefficient $\gamma$ | 0.9 |
| **3**
 **(single flow channel, SFDM)** | $Q, m, T_0, Pe$ | same as Test 1 |
| | Diffusion parameter $\beta$ | 0.04 h$^{-1/2}$ |
| **4**
 **(single flow channel, 2RNE)** | $Q, m, T_0, Pe$ | same as Test 1 |
| | Length of the flow channel $L$ | 1000 m |
| | Fraction of mobile water $\psi$ | 0.7 |
| | Omega coefficient $\omega$ | 0.1 m$^{-1}$ |
| **5**
 **(two channels, MDM-ADE)** | Total system flow rate $Q$ | 10 m$^3$ h$^{-1}$ |
| | Mass flowing through the first channel $m_1$ | 12 g |
| | Mass flowing through the second channel $m_2$ | 8 g |
| | Mean transit time in the first channel $T_{01}$ | 170 h |
| | Mean transit time in the second channel $T_{02}$ | 300 h |
| | Peclet number in the first channel $Pe_1$ | 15 |
| | Peclet number in the second channel $Pe_2$ | 80 |

[Figure]

**Figure 2.** Comparison among MFIT, CXTFIT, and TRACI simulations for test 1 (single flow channel, ADE, instantaneous injection), test 2 (single flow channel, ADE, exponentially decaying injection), test 3 (single flow channel, SFDM), test 4 (single flow channel, 2RNE), and test 5 (two channels, MDM-ADE)

*Regarding the modeled BTCs from the HES experimental site, they also display short tracer test duration and local transport dynamics. Some questions arise me, what about longer –multi-kilometers- karst connections and their expected very often long-tailed BTCs? and, what about the degree of flow diversion in anastomosed/ forked karst conduit systems and their associated multi-peak BTCs? I agree with the proposed pathway decomposition in multi-single channel scheme but, how the flow diversion in one or several of them and where (close to the injection point or to the end of the master conduit) may condition the obtained BTC shape?*

There is no specific scale attached to the multiflow modeling approach, as depicted in Fig. 1. The lengths of the flow channels can be assumed to be a few meters or several kilometers. The proposed model is applicable to any tracer test, regardless of the distance between the injection site and the monitoring point. Likewise, short-, medium-, or long-duration BTCs can be simulated with the model depending on the values of the parameters that influence the spreading of transit/residence times in the individual flow channels ($Pe$, $\gamma$, $\beta$, $\psi$, and $\omega$) and/or by considering multiple flow channels that have a large span of mean transit time ($T_0$) values. The only unsuitable cases are those for which extensive breakthrough tailing would be due to i) unsteady flow, and/or ii) complex injection signals (e.g., multiple steps), and/or iii) reactive transport processes, which are not considered in the present model version. A noticeable difference between modeling short- or long-distance tracer tests could be the physical interpretation that pertains to the channels of the multiflow model. For short distances, the channels might be considered as model-abstractions of real individual fractures or karst-conduits. For longer distances, this abstraction likely would be physically unrealistic. As discussed in the manuscript (L105−108), in the general case, the channels are not assumed to represent individual fractures or karst conduits but are lumped submodels of the main flow routes employed by the tracer through the fractures/karst conduit network. The multiflow model that will be fitted against a multi-peak BTC generated by a complex flow structure geometry will always be a simplification of the real flow network pattern, which is the essence of modeling. Moreover, the information content of a tracer BTC only does not allow for further structural interpretation. For

instance, the existence of two concentration peaks in a tracer BTC indicates at least one diverging-converging flow structure between the injection site and the monitoring point. However, many more structures may exist in reality; their effects can be masked by mixing at the converging nodes and/or by similar transit times in the different pathways. Similarly, it is impossible to determine the relative location of the diverging and converging nodes between the injection site and the monitoring point. Promising approaches to address this issue are i) the joint inversion of multiple tracer tests with different injection and monitoring locations, which is also referred to as tracer tomography (see, e.g., Borghi et al., Can one identify karst conduit networks geometry and properties from hydraulic and tracer test data? Adv. Water Resour. 2016) and ii) the joint interpretation of tracer test and geophysical data (see, e.g., Shakas et al., Probabilistic inference of fracture-scale flow paths and aperture distribution from hydrogeophysically-monitored tracer tests, J. Hydrol. 2018).

*I miss complementary numerical results such as transport parameters and their discussion (i.e. sensitivity analysis) for a deeper comparative analysis of simulation results in section 5. The recovery rate of the injected tracer for the three examined BTCs would be helpful to the reader to have information about how many tracer mass has been lost during the test. This will help to understand the potential role of rock matrix or stagnant zones in the karst circuit by which anomalous transport is reflected as multi-peaks or long-tailed BTC shapes.*

A table that lists the fitted parameter values that correspond to Fig. 4 has been prepared: https://doi.org/10.5281/zenodo.3824439. The table also contains i) the injected and recovered mass values for each tracer experiment, ii) the corresponding recovery ratios, iii) the minimum and maximum parameter values considered for the optimization, iv) the composite parameter sensitivities computed with PEST, and v) the post-calibration scaled values of the parameters pairs $(Q, m_j)$ and $(C_0, Q_j/Q)$, which are consistent with the recovered mass values. This post-calibration scaling was necessary because no parameter was fixed prior to the inversion; so a degree of freedom remained in the pairs of parameters due to their balance in the model equations. Owing to its large size (more than 400 parameters for the different models and channel number scenarios), I propose the inclusion of this table as a supplement rather than as an appendix. I also added a discussion about the (consistent) variation in the values of the parameters that influence the spreading of transit/residence times in the individual flow channels according to the different models and/or the number of flow channels.

The optimized parameter values and their composite sensitivities at the end of the optimization process are provided in the Supplement (Table S1). Unsurprisingly, the model parameters that influence the spreading of transit/residence times in the individual flow channels while accounting for different processes ($Pe$, $\gamma$, $\beta$, $\psi$, and $\omega$) are sensitive to the number of channels. For instance, when comparing single- with multiple-channel models, the former requires lower $Pe$ values to compensate for the coarser description of the flow system heterogeneity (recall that the dispersion coefficient integrated in the Peclet number reflects the unresolved variability of the flow velocity below the modeling scale). The same observation holds when comparing single- and double-porosity models with the same number of flow channels, i.e., the $Pe$ values of single-porosity models are lower than the $Pe$ values of double-porosity models because part of the spreading of transit/residence times in the latter case is implicitly captured by solute mass exchanges between the mobile and immobile domains. A noticeable exception is the diffusion parameter $\beta$ of the SFDM model, whose values are mostly around $1.0 \times 10^{-3}$ h$^{-1/2}$. This value corresponds to the upper bound of the optimization range set for this parameter, which is based on a matrix porosity of 30 %, a molecular diffusion coefficient of $1.0 \times 10^{-9}$ m$^2$ s$^{-1}$, and a flow-channel aperture of $1.0 \times 10^{-2}$ m. Beta values larger than $1.0 \times 10^{-3}$ h$^{-1/2}$ would be physically unrealistic. The fact that the Beta value is limited by its upper bound during the optimization process indicates that the SFDM model is not suitable for describing the HES tracer experiments, as further discussed below. All other parameters have converged to values far from their optimization bounds.

Concerning your last remark, it is not possible, with the specified assumptions of steady-state flow and non-reactive tracer, to establish a link between the incomplete recovery of a tracer and mass-exchanges between flowing and stagnant water regions. These exchanges are assumed to obey physically reversible processes (regarded either as a first-order system or second-order system in the 2RNE model and SFM model, respectively), and therefore, cannot be regarded as potentially responsible for the incomplete mass recovery. For tracer tests performed in steady state conditions that involve non-reactive tracers, an incomplete recovery of the injected mass indicates a diverging flow structure between the injection site and the monitoring point. Unfortunately, no additional information can be drawn about this flow divergence from the recovered tracer data only. This discussion has been added to the revised manuscript.

The mass recovery ratios for these three tracer experiments were 58 %, 79 %, and 60 %, respectively. Note that these recovery data cannot be included in the model because the flow structure assumption that underlies the multiflow approach (Fig. 1) implies that all the mass that enters the system flows out after a certain lapse of time. The same holds for any single- or double-porosity modeling approach based on a 1-D flow assumption. For tracer tests that are performed in steady state conditions and involve non-reactive tracers, an incomplete recovery of the injected mass indicates a diverging flow structure between the injection site and the monitoring point. Unfortunately, no additional information can be obtained about this flow divergence from the tracer data only. Therefore, the total mass in a multiflow model must be consistent with the recovered tracer mass rather than the injected mass.

*I recommend adding at the early sections a glossary of acronyms and parameters described throughout the text.*

The acronyms and model parameters have been summarized in two separate tables. Owing to their size, I felt that it was more appropriate to place these two tables (A1 and A2) in the Appendix. Two sentences have been added at the end of the Introduction to refer to these tables. Four new references quoted in Table A1 (Diersch, 2014; Langevin et al., 2017; Zheng et al., 2012; Zheng and Bennett, 2002) have been added to the References.

[revised manuscript text omitted]

_Point-to-point comments:_

_Page 12: Table 2, test 4 >>>  "Partitioning coefficient (ß)" instead of "Fraction of mobile water (ψ)"?_

_Page 12: Table 2, test 4 >>>  "Mass transfer coefficient" instead of "Omega coefficient"?_

As mentioned in Table 2 and the text (lines 296–298), test 4 addresses the case of a single-flow channel that is described as a two-region nonequilibrium (2RNE) medium. The 2RNE model is mathematically described by Equations (13) – (17) and the

model parameters are summarized in Table 1. The parameters $\psi$ and $\omega$ are part of the 2RNE model and correct in Table 2, test 4. The suggested changes would be inappropriate since the parameter $\beta$ is the diffusion parameter in the SFDM model, and the "mass transfer coefficient" corresponds more specifically to the parameter $\alpha$, which is part of $\omega$ (refer to line 175 and Eq. 17).

**Anonymous Referee #2**

*Figure 1: should this say what the dashed line represents, is it non flowing water?*

The dashed line was supposed to represent possible additional flow channels (greater than 3 and less than *N*). I removed this line for clarity.

*Line 115-116: "A possible reason is the increasing number of fitting parameters, which makes the inverse problem more complicated. The use of modern inversion tools such as PEST enables overcoming this problem, as discussed in section 3" I agree that these methods can efficiently find parameters sets in the situation you outline but I would assume not without the possibility that the parameters best fitting the data are far from unique and more so the greater the number of parameters. For me, this sentence misses a discussion of this important caveat in an otherwise very carefully considered section.*

A short discussion about the non-uniqueness (equifinality) issue has been added in this section.

Among the challenges related to the inversion of a multiflow model is the inherent problem of nonuniqueness (or equifinality). A variety of parameter sets can yield nearly identical simulated BTCs because the change in the value of a parameter of a given channel can be compensated by modifying at least one other parameter that pertains to this same channel or the parameters of the other channels. This nonuniqueness causes the inverse problem to be ill-posed in the sense of Hadamard (1902) and requires the use of advanced optimization methods, such as regularization, to make the inverse problem tractable (Tikhonov and Arsenin, 1977; Moore and Doherty, 2006; Zhou et al., 2014).

The uncertainty associated with the inverted parameter set and the methods that can be employed to assess this uncertainty are addressed again at the end of Chapter 3 and at the end of section 5 (application example).

*Around line 235: For my understanding, is the optimisation run for a given number of channels and if so should the user seek the minimum number of n that perform well for the measurement objective function and regularisation terms. OK, I see later where this comes in but I'll leave the comment so you can see the issue I had when reading for the first time.*

Clarification was necessary; the text has been amended.

The optimization and uncertainty analysis of the model parameters for a given number of flow channels are carried out using PEST routines (Doherty, 2019a, 2019b). The influence of the number of channels on the model fitting performance can be analyzed once a series of calibrations has been performed for a variety of channel numbers, as illustrated below.

*Around line 285: A series of utility functions are called here for the uncertainty analysis. I don't think they need further explanation here but a pointer to the relevant documentation/literature on these would aid completeness.*

A sentence that specifies appropriate references has been added.

The method is essentially similar to that described by Fang et al. (2019) and relies on the use of the PREDUNC7 and RANDPAR utilities documented in the PEST manual (Doherty, 2019b).

*Section 4 code verification – should this also test for the case where n channels is unknown? So for test 5 if n_max was set to 6 would the same results be found as for the current test. Perhaps this goes beyond verification of the transport processes models, which is clearly the aim of this section, but I think checking the multistart would add value if feasible.*

A new test has been implemented to assess the multistart method. The related discussion has been placed in a new/separate section (section 5) for the sake of clarity.

**5 Assessment of the multistart optimization method**

The purpose of this section is to assess the automatic multistart method described in section 3 using a new synthetic test case. A multimodal BTC that corresponds to 3 channels has been simulated using the MDMi program with the parameters listed in Table 3. A "blind" inversion of this BTC has been performed using the automatic multistart method with a maximum number of flow channels $N_{max} = 6$. The only model parameter that has been fixed prior to the inversion process was the total flow rate $Q$ to simplify the post-comparison of the inverted mass values in each channel with the "true" mass values. Otherwise, a degree of freedom would persist for the pairs of the optimized $Q$ and $m_j$ values, i.e., multiplying or dividing these parameters by the same constant would yield the same BTCs; refer to Eq. (6). The parameters $m_j$, $T_{0j}$ and $Pe_j$ of the different flow channels were optimized with virtually no upper and lower bound constraints (minimum and maximum allowed parameter values of $1.0 \times 10^{-10}$ and $1.0 \times 10^{+10}$, respectively). As shown in Fig. 3, the inverted BTCs that correspond to $N = 3, 4, 5,$ and $6$ channels overlap perfectly with each other and with the original simulated BTC; and as shown in Table 4, the optimized values for the parameters of the 3-channel model are equal to the "true" parameter values.

**Table 3.** Model parameters that correspond to the multimodal simulated BTC in Fig. 3.

| Parameters | Values |
|:---:|:---:|
| $Q$ | 10 m$^3$ h$^{-1}$ |
| $m_1$ | 10 g |
| $m_2$ | 6 g |
| $m_3$ | 4 g |
| $T_{01}$ | 150 h |
| $T_{02}$ | 250 h |
| $T_{03}$ | 350 h |
| $Pe_1$ | 20 |
| $Pe_2$ | 50 |
| $Pe_3$ | 100 |

[Figure]

**Figure 3.** Inversion of the 3-channel-simulated BTC using the automatic multistart method with $N_{max} = 6$. The inverted BTCs that correspond to $N = 3, 4, 5,$ and 6 channels overlap perfectly with each other and the original simulated BTC.

**Table 4.** Optimized model parameters that correspond to the inverted BTCs in Fig. 3.

| $N$ | 1 | 2 | 3 | 4 | 5 | 6 |
|---|---|---|---|---|---|---|
| $m_1$ (g) | 21.11 | 10.79 | 10.00 | 2.79 | 2.66 | 2.66 |
| $m_2$ (g) | - | 9.54 | 6.00 | 7.19 | 7.35 | 7.35 |
| $m_3$ (g) | - | - | 4.00 | 6.02 | 5.99 | 5.91 |
| $m_4$ (g) | - | - | - | 4.00 | 2.58 | 2.62 |
| $m_5$ (g) | - | - | - | - | 1.42 | 1.45 |
| $m_6$ (g) | - | - | - | - | - | 0.02 |
| $T_{01}$ (h) | 239.36 | 155.82 | 150.00 | 126.17 | 151.55 | 151.31 |
| $T_{02}$ (h) | - | 302.91 | 250.00 | 158.91 | 149.47 | 149.60 |
| $T_{03}$ (h) | - | - | 350.00 | 250.00 | 249.97 | 249.30 |
| $T_{04}$ (h) | - | - | - | 350.01 | 349.48 | 347.68 |
| $T_{05}$ (h) | - | - | - | - | 350.84 | 351.08 |
| $T_{06}$ (h) | - | - | - | - | - | 405.58 |
| $Pe_1$ | 6.72 | 17.52 | 20.00 | 24.22 | 19.80 | 19.92 |
| $Pe_2$ | - | 27.55 | 50.00 | 22.18 | 20.07 | 20.01 |
| $Pe_3$ | - | - | 100.00 | 49.94 | 50.04 | 50.62 |
| $Pe_4$ | - | - | - | 100.00 | 98.45 | 88.42 |
| $Pe_5$ | - | - | - | - | 102.68 | 120.27 |
| $Pe_6$ | - | - | - | - | - | 442.13 |

*Figure 3: In my version the dots and labels overlap, generally this figure could be cleaner, and the scale bar is also quite small. Would it also be possible to highlight the wells used for pumping and injections in the experiments, perhaps with colours or different symbology.*

This figure (Fig. 4 in the revised manuscript) has been revised.

[Figure]

**Figure 4.** Locations of wells at the HES in Poitiers, France. Map data are from Google.

*Line 345: Do the parameter bounds come into play in the optimised parameter sets? i.e. do you get parameters optimising to the bounds? Generally, there is not any discussion of the parameters found, we there a reason for this? I think this should be justified.*

A table that lists the fitted parameter values that correspond to Fig. 4 has been prepared: https://doi.org/10.5281/zenodo.3824439. The table also contains i) the injected and recovered mass values for each tracer experiment, ii) the corresponding recovery ratios, iii) the minimum and maximum parameter values considered for the optimization, iv) the composite parameter sensitivities computed with PEST, and v) the post-calibration scaled values of the parameters pairs $(Q, m_j)$ and $(C_0, Q_j/Q)$, which are consistent with the recovered mass values. This post-calibration scaling was necessary because no parameter was fixed prior to the inversion; so a degree of freedom remained in the pairs of parameters due to their balance in the model equations. Owing to its large size (more than 400 parameters for the different models and channel number scenarios), I propose the inclusion of this table as a supplement rather than an appendix. I also added a discussion about the (consistent) variation in the values of the parameters that influence the spreading of transit/residence times in the individual flow channels according to the different models and/or number of flow channels. A bound effect on the optimization of parameter $\beta$ (SFDM model) has been noted and discussed.

The optimized parameter values and their composite sensitivities at the end of the optimization process are provided in the Supplement (Table S1). Unsurprisingly, the model parameters that influence the spreading of transit/residence times in the individual flow channels while accounting for different processes ($Pe$, $\gamma$, $\beta$, $\psi$, and $\omega$) are sensitive to the number of channels. For instance, when comparing single- with multiple-channel models, the former requires lower $Pe$ values to compensate for the coarser description of the flow system heterogeneity (recall that the dispersion coefficient integrated in the Peclet number reflects the unresolved variability of the flow velocity below the modeling scale). The same observation holds when comparing single- and double-porosity models with the same number of flow channels, i.e., the $Pe$ values of single-porosity models are lower than the $Pe$ values of double-porosity models because part of the spreading of transit/residence times in the latter case is implicitly captured by solute mass exchanges between the mobile and immobile domains. A noticeable exception is the diffusion parameter $\beta$ of the SFDM model, whose values are mostly around $1.0 \times 10^{-3}$ h$^{-1/2}$. This value corresponds to the upper bound of the optimization range set for this parameter, which is based on a matrix porosity of 30 %, a molecular diffusion coefficient of $1.0 \times 10^{-9}$ m$^2$ s$^{-1}$, and a flow-channel aperture of $1.0 \times 10^{-2}$ m. Beta values larger than $1.0 \times 10^{-3}$ h$^{-1/2}$ would be physically unrealistic. The fact that the Beta value is limited by its upper bound during the optimization process indicates that the SFDM model is not suitable for describing the HES tracer experiments, as further discussed below. All other parameters have converged to values far from their optimization bounds.

*Figure 5: Could the legend be a single legend for all plots. On my version the legend is also quite small making dashed and continuous lines difficult to identify. Worth checking in the final production of the figure for publication.*

The figure has been modified to include only one legend. The readability problems are most likely related to the .pdf file that was generated for the review. This figure is a vector graphic and should not pose any production/re-editing issues if the article is accepted for publication in GMD.

*Line 390 uncertainty analysis – Could you be more explicit about why the particular model and test case was chosen for the uncertainty analysis. Line 390 uncertainty analysis – Could you be more explicit about why the particular model and test case was chosen for the uncertainty analysis. Line 396: "fairly similar" could you be more precise about how similar was defined. The uncertainty analysis description in the methods is quite brief which means its difficult to fully appreciate the setup here in my opinion. Around line 400 – I feel the discussion of these results is somewhat rushed regarding the uncertainty analysis, I don't feel I fully appreciate the results. Is this conclusion made because only the four and six channel models capture the first peak? MDP-2RNE seems to for the two channels although it's difficult to see if this is really the case.*

Since this part of the original manuscript was unclear, I rewrote it and expanded the uncertainty analysis by considering i) two models (MDMi and MDP-2RNE) instead of one model (MDMi in the original manuscript) and ii) different numbers of channels (1, 2, and 3) in each case instead of a single two-channel model in the original manuscript. As outlined in the revised manuscript, the objective is to illustrate the possibilities offered by MFIT for the analysis of uncertainties associated with the inverted parameters. It would be inappropriate to rely on this analysis to compare the performances of the different models since, as discussed in the revised manuscript, the choice of a model and a number of channels (i.e., degree of complexity of the model) depends on the objective pursued by the modeler. In the revised manuscript, the results of Fig. 7 and Fig. 8 are discussed in terms of the equifinality of the inverted model parameters.

[revised manuscript text omitted]